# LEARNING TO DISSIPATE ENERGY IN OSCILLATORY STATE-SPACE MODELS

## ABSTRACT

State-space models (SSMs) are a class of networks for sequence learning that benefit from fixed state size and linear complexity with respect to sequence length, contrasting the quadratic scaling of typical attention mechanisms. Inspired from observations in neuroscience, Linear Oscillatory State-Space models (LinOSS) are a recently proposed class of SSMs constructed from layers of discretized forced harmonic oscillators. Although these models perform competitively, leveraging fast parallel scans over diagonal recurrent matrices and achieving state-of-the-art performance on tasks with sequence length up to 50k, LinOSS models rely on rigid energy dissipation ("forgetting") mechanisms that are inherently coupled to the time scale of state evolution. As forgetting is a crucial mechanism for long-range reasoning, we demonstrate the representational limitations of these models and introduce Damped Linear Oscillatory State-Space models (D-LinOSS), a more general class of oscillatory SSMs that learn to dissipate latent state energy on arbitrary time scales. We analyze the spectral distribution of the model's recurrent matrices and prove that the SSM layers exhibit stable dynamics under a simple, flexible parameterization. Without additional complexity, D-LinOSS consistently outperforms previous LinOSS methods on long-range learning tasks, achieves faster convergence, and relinquishes the need for multiple discretization schemes.

## 1 INTRODUCTION

State-space models (SSMs) (Gu et al., 2021; Smith et al., 2023; Gu & Dao, 2023; Hasani et al., 2022; Rusch & Rus, 2025) have emerged as a powerful deep learning architecture for sequence modeling, demonstrating strong performances across various domains, including natural language processing (Gu & Dao, 2023), audio generation (Goel et al., 2022), reinforcement learning (Lu et al., 2024), and scientific and engineering applications (Hu et al., 2024).

Despite the abundance of neural network architectures for sequence modeling, SSMs have gained significant attention due to their fundamental advantages over both Recurrent Neural Networks (RNNs) and Transformer architectures based on self-attention mechanisms (Vaswani, 2017). Built upon layers of sequence-to-sequence transformations defined by linear dynamical systems, SSMs integrate principles from control theory with modern deep learning techniques, making them highly effective across multiple modalities. While recent SSM architectures are often formulated as linear RNNs (Orvieto et al., 2023), they introduce notable improvements over their predecessors, offering enhanced speed, accuracy, and the ability to capture long-range dependencies more effectively.

In this work, we focus on the regime of linear, time-invariant (LTI) SSMs and extend the recently introduced Linear Oscillatory State-Space model (LinOSS) (Rusch & Rus, 2025). LinOSS formulates a continuous-time system of second-order ordinary differential equations (ODEs) that represent forced harmonic oscillators. These dynamics are then discretized into two conditionally stable state-space variants, each derived from a different ODE integration method, and the resulting models are computed efficiently by leveraging associative parallel scans. The structure of the underlying oscillatory dynamics allows LinOSS to learn long-range interactions with minimal constraints on the SSM state matrix. However, previous LinOSS models inherently couple the frequency and damping behaviors of state-space layers, effectively collapsing latent state energy dissipation to a single scale and limiting the model's expressivity. To overcome this, we introduce a flexible and controllable

**LinOSS** single scale damping

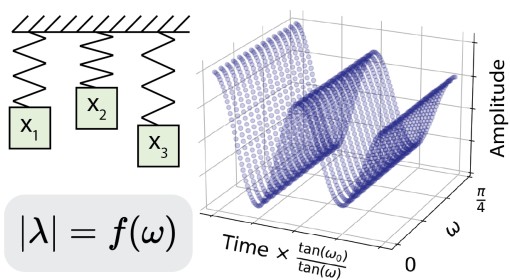

$$|\lambda| = f(\omega)$$

**Damped LinOSS** multi-scale damping

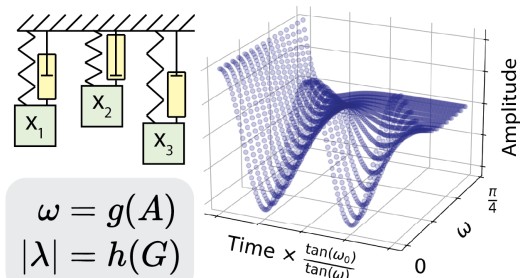

$$\omega = g(A)$$
$$|\lambda| = h(G)$$

Figure 1: Previous LinOSS models are derived from *harmonic oscillators*, directly coupling the frequency $\omega$ and magnitude $|\lambda|$ of discretized eigenvalues and reducing state energy dissipation to a single scale when normalizing time by frequency. Instead derived from *damped harmonic oscillators*, **D-LinOSS** learns $\omega$ and $|\lambda|$ independently. The range of representable second-order systems within each underlying state-space layer is shown above; the particular relationship between frequency and damping depicted in the right diagram can be selected arbitrarily.

dissipation mechanism and derive the *Damped Linear Oscillatory State-Space Model (D-LinOSS)*, enhancing the LinOSS architecture by incorporating learnable damping independent of time scale.

Our approach constructs a deep state space model capable of capturing a wide range of temporal relationships by expanding the expressivity of individual SSM layers. Unlike previous versions of LinOSS that were constrained to a limited subset of oscillatory systems, our method allows each layer to independently learn a wider range of stable oscillatory dynamics, collectively leading to a more powerful sequence model. Our full contributions are:

- We conduct a rigorous spectral analysis of the proposed D-LinOSS model, highlighting the representational improvements enabled by learnable damping.
- We validate the theoretical expressivity improvements through a synthetic experiment of learning exponential decay.
- We derive a stable parameterization of D-LinOSS and introduce an initialization procedure to generate arbitrary eigenvalue distributions in the recurrent matrix. We perform ablations comparing different initialization techniques.
- We provide extensive empirical evaluation, showing that D-LinOSS on average outperforms state-of-the-art models across eight different challenging real-world sequential datasets.
- We showcase the additional practical benefits of D-LinOSS, such as faster convergence and a smaller hyperparameter space by eliminating the need for multiple discretization schemes.

## 2 BACKGROUND

### 2.1 CONTINUOUS-TIME FORMULATION

D-LinOSS layers are constructed from a system of damped, forced harmonic oscillators:

$$\mathbf{y}''(t) = -\mathbf{A}\mathbf{y}(t) - \mathbf{G}\mathbf{y}'(t) + \mathbf{B}\mathbf{u}(t),$$
$$\mathbf{x}(t) = \mathbf{C}\mathbf{y}(t) + \mathbf{D}\mathbf{u}(t) \tag{1}$$

The continuous-time parameters $\mathbf{A}$ and $\mathbf{G}$ are restricted to diagonal matrices with non-negative entries, meaning (1) is an uncoupled second-order system. The feed-forward operation $\mathbf{D}\mathbf{u}(t)$ will be omitted for the rest of the paper for concision.

D-LinOSS layers provide an expressive, stable, and efficient recurrent primitive for modeling intermediate sequence-to-sequence transformations $\mathbf{u} \mapsto \mathbf{x}$ in $\mathbb{R}^m$ through learning parameters $\mathbf{A}$, $\mathbf{G}$,

**B**, and **C**. **A** controls the natural frequency of the system's oscillation and **G** defines the damping, i.e., the energy dissipation of the latent state. The underlying dynamical system of previous LinOSS models is (1) subject to $\mathbf{G} = \mathbf{0}$; thus, D-LinOSS is constructed from a more general oscillatory dynamical system with learnable damping. The additional $m$ learnable parameters from **G** are a negligible contribution to model size and have no impact on speed.

## 2.2 DISCRETIZATION

The D-LinOSS discrete-time state-space layer is derived by adopting an integration scheme to approximately solve System (1) as an initial-value problem (IVP) subject to $\mathbf{y}(0) = \mathbf{y}'(0) = \mathbf{0}$. This discretization technique is effectively a specification for mapping the underlying continuous-time parameters **A**, **G**, and **B** to discrete-time counterparts **M** and **F** in the following systems.

$$\begin{bmatrix} \mathbf{z}'(t) \\ \mathbf{y}'(t) \end{bmatrix} = \begin{bmatrix} -\mathbf{G} & -\mathbf{A} \\ \mathbf{I} & \mathbf{0} \end{bmatrix} \begin{bmatrix} \mathbf{z}(t) \\ \mathbf{y}(t) \end{bmatrix} + \begin{bmatrix} \mathbf{B} \\ \mathbf{0} \end{bmatrix} \mathbf{u}(t) \quad (2) \quad \longrightarrow \quad \begin{bmatrix} \mathbf{z}_{k+1} \\ \mathbf{y}_{k+1} \end{bmatrix} = \mathbf{M} \begin{bmatrix} \mathbf{z}_k \\ \mathbf{y}_k \end{bmatrix} + \mathbf{F}\mathbf{u}_{k+1} \quad (3)$$

Discretization also introduces learnable time-step parameters $\Delta t \in \mathbb{R}^m$ that govern the integration interval for the ODE solution.

Unlike standard first-order SSMs, oscillatory SSMs explicitly model the acceleration and velocity of the system state, resulting in smoother outputs due to the twice-integrated dynamical structure. As a result, although most SSMs discretize the continuous-time dynamics using zero-order hold or the bilinear method, the second-order structure of D-LinOSS necessitates the use of special discretization schemes to maintain conditional system stability without over-constraining the matrices **A** and **G**.

Specifically, Rusch & Rus (2025) investigate the use of implicit integration (IM) and symplectic integration, also referred to as implicit-explicit integration (IMEX), as discretization methods for their proposed continuous-time system of *undamped* harmonic oscillators. Each integrator endows the resulting SSM with different energy dissipation properties; IM integration produces a dissipation term coupled to the eigenvalue phase and IMEX integration completely preserves energy across time. The selection of discretization technique is thus a binary hyperparameter in the original LinOSS model used to modulate the amount of "forgetting," giving rise to two SSMs (LinOSS-IM and LinOSS-IMEX) exhibiting different dynamical behaviors.

We extend the use of IMEX integration to the D-LinOSS framework, as learnable damping allows for full dynamical control regardless of which discretization method is used. This flexibility in parameterization removes the need to treat discretization scheme as a binary hyperparameter, reducing the model search space.

Applying IMEX integration to System (1) yields:

$$\begin{aligned} \mathbf{z}_{k+1} &= \mathbf{z}_k + \Delta t\big(-\mathbf{A}\mathbf{y}_k - \mathbf{G}\mathbf{z}_{k+1} + \mathbf{B}\mathbf{u}_{k+1}\big), \\ \mathbf{y}_{k+1} &= \mathbf{y}_k + \Delta t\mathbf{z}_{k+1} \end{aligned} \quad (4)$$

or in matrix form,

$$\begin{bmatrix} \mathbf{I} + \Delta t\mathbf{G} & \mathbf{0} \\ -\Delta t\mathbf{I} & \mathbf{I} \end{bmatrix} \begin{bmatrix} \mathbf{z}_{k+1} \\ \mathbf{y}_{k+1} \end{bmatrix} = \begin{bmatrix} \mathbf{I} & -\Delta t\mathbf{A} \\ \mathbf{0} & \mathbf{I} \end{bmatrix} \begin{bmatrix} \mathbf{z}_k \\ \mathbf{y}_k \end{bmatrix} + \begin{bmatrix} \Delta t\mathbf{B} \\ \mathbf{0} \end{bmatrix} \mathbf{u}_k \quad (5)$$

Inverting the left hand side block matrix, we arrive at the final discrete-time SSM in the form of (3).

$$\mathbf{M} := \begin{bmatrix} \mathbf{S}^{-1} & -\Delta t\mathbf{S}^{-1}\mathbf{A} \\ \Delta t\mathbf{S}^{-1} & \mathbf{I} - \Delta t^2\mathbf{S}^{-1}\mathbf{A} \end{bmatrix}, \quad \mathbf{F} := \begin{bmatrix} \Delta t\mathbf{S}^{-1}\mathbf{B} \\ \Delta t^2\mathbf{S}^{-1}\mathbf{B} \end{bmatrix} \quad (6)$$

Here, the Schur complement is the diagonal matrix $\mathbf{S} = \mathbf{I} + \Delta t\mathbf{G}$ and **M** is a block matrix composed of diagonal sub-matrices.

## 2.3 ASSOCIATIVE PARALLEL SCANS

Many modern SSM architectures (Smith et al., 2023) leverage associative parallel scans (Kogge & Stone, 1973; Blelloch, 1990) to efficiently compute recurrent operations across long sequences. By exploiting the associativity of the recurrence operator, naively $O(N)$ sequential operations can be parallelized and computed in $O(\log N)$ time. For SSMs, parallel scans enable sub-linear complexity of the recurrence computation with respect to sequence length, acting as a key building block for scaling SSMs to long contexts.

## 3 THEORETICAL PROPERTIES

Spectral analysis provides a lens to examine the stability and dynamical behavior of SSMs. In the absence of bounding nonlinearities like $\tanh$, the eigenvalues of the recurrent matrix $\mathbf{M}$ fully govern how latent states evolve across time. In particular, eigenvalues with near unit norm retain energy across long time horizons, while those closer to zero rapidly dissipate energy.

In the previous LinOSS-IM and LinOSS-IMEX models, the internal system spectra are rigidly defined by the selection of discretization technique, coupling frequency and damping. As shown in Figure 1, this effectively reduces latent state energy dissipation to a single scale, limiting the range of expressible dynamics. For D-LinOSS, the spectrum of $\mathbf{M}$ instead arises from damped harmonic oscillators, introducing a new tunable mechanism that decouples damping from frequency. Unlike the preceding models, D-LinOSS layers can represent all stable second-order systems, yielding a broader range of expressible dynamics and thus a more powerful sequence model. Figure 1 depicts this, where the scale of energy dissipation can be arbitrarily selected regardless of oscillation frequency.

These notions are formalized in this section, where we characterize the eigenvalues of D-LinOSS, derive stability conditions, and compare the resulting spectral range to that of previous LinOSS models. In particular, we show that the set of reachable, stable eigenvalue configurations in D-LinOSS is the full complex unit disk, where that of LinOSS has zero measure in $\mathbb{C}$.

### 3.1 SPECTRAL ANALYSIS AND STABILITY

**Proposition 3.1.** *The eigenvalues of the D-LinOSS recurrent matrix $\mathbf{M} \in \mathbb{R}^{2m \times 2m}$ are*

$$\lambda_{i_{1,2}} = \frac{1 + \frac{\Delta t_i}{2}\mathbf{G}_i - \frac{\Delta t_i^2}{2}\mathbf{A}_i}{1 + \Delta t_i \mathbf{G}_i} \pm \frac{\frac{\Delta t_i}{2}\sqrt{(\mathbf{G}_i - \Delta t_i \mathbf{A}_i)^2 - 4\mathbf{A}_i}}{1 + \Delta t_i \mathbf{G}_i}, \tag{7}$$

*where pairs of eigenvalues are denoted as $\lambda_{i_{1,2}}$ and $i = 1, 2, ..., m$.*

*Proof.* The derivation is provided in Appendix A.1. Because the $m$ second-order systems are decoupled, it is sufficient to subsequently analyze the spectral properties and stability conditions for a single system with index $i \in 1, \ldots, m$. $\square$

Proposition 3.1 shows that eigenvalues of D-LinOSS are tuned through choices of $\mathbf{A}$, $\mathbf{G}$, and $\Delta t$. We now detail a sufficient condition for system stability.

**Proposition 3.2.** *Assume $\mathbf{A}_i$, $\mathbf{G}_i$ are non-negative, and $\Delta t_i \in (0, 1]$. If the following is satisfied:*

$$(\mathbf{G}_i - \Delta t_i \mathbf{A}_i)^2 \le 4\mathbf{A}_i, \tag{8}$$

*then $\lambda_{i_{1,2}}$ come in complex conjugate pairs $\lambda_i$, $\lambda_i^*$ with the following magnitude:*

$$|\lambda_i| = \frac{1}{\sqrt{1 + \Delta t_i \mathbf{G}_i}} \le 1, \tag{9}$$

*i.e., the eigenvalues are unit-bounded. Define $\mathcal{S}_i$ to be the set of all $(\mathbf{A}_i, \mathbf{G}_i)$ that satisfy the above condition. For notational convenience, we order the eigenvalues such that $\mathrm{Im}(\lambda_i) \ge 0$, $\mathrm{Im}(\lambda_i^*) \le 0$.*

*Proof.* The proof is detailed in Appendix A.2. Condition (8) is simply the non-positivity of the discriminant in the eigenvalue expression of Proposition 3.1, which is shown to be sufficient for the unit-boundedness of $|\lambda_i|$. □

We now demonstrate that the spectral image of $\mathcal{S}_i$ is the full unit disk, meaning D-LinOSS is capable of representing every stable, damped, uncoupled second-order system.

**Proposition 3.3.** *The mapping* $\Phi : \mathcal{S}_i \to \mathbb{C}_{|z|\leq 1} \setminus \{0\}$ *defined by* $(\mathbf{A}_i, \mathbf{G}_i) \mapsto (\lambda_i, \lambda_i^*)$ *is bijective.*

*Proof.* In Appendix A.3, a well-defined inverse mapping $\Phi^{-1} : \mathbb{C}_{|z|\leq 1} \setminus \{0\} \to \mathcal{S}_i, \ (\lambda_i, \lambda_i^*) \mapsto (\mathbf{A}_i, \mathbf{G}_i)$ is constructed. This inverse map has practical utility in matrix initialization, enabling the selection of arbitrary distributions of stable eigenvalues. □

In contrast to the full expressive range of D-LinOSS layers, LinOSS-IM and LinOSS-IMEX layers are limited in reachable eigenvalues. We recall the respective expressions from Rusch & Rus (2025):

$$\lambda_{i_{1,2}}^{\text{IM}} = \frac{1}{1 + \Delta t_i^2 \mathbf{A}_i} \pm j \frac{\Delta t_i \sqrt{\mathbf{A}_i}}{1 + \Delta t_i^2 \mathbf{A}_i}, \quad \lambda_{i_{1,2}}^{\text{IMEX}} = \frac{1}{2}(2 - \Delta t_i^2 \mathbf{A}_i) \pm \frac{j}{2}\sqrt{\Delta t_i^2 \mathbf{A}_i(4 - \Delta t_i^2 \mathbf{A}_i)}, \quad (10)$$

It can be seen that both forms impose a rigid relationship between frequency and damping, constraining the reachable spectra. The following proposition formalizes this by showing that the set of eigenvalues reachable under these parameterizations occupies zero area within the unit disk. Figure 2 shows the respective expressions graphed on $\mathbb{C}$ for a visual interpretation.

**Proposition 3.4.** *For both LinOSS-IM and LinOSS-IMEX, the set of eigenvalues constructed from* $\mathbf{A}_i \in \mathbb{R}_{\geq 0}$ *and* $\Delta t_i \in (0, 1]$ *is of measure zero in* $\mathbb{C}$.

*Proof.* Detailed in Appendix A.4, the change of variables $\gamma_i = \Delta t_i \sqrt{\mathbf{A}_i}$ renders both eigenvalue expressions one-dimensional curves, which have zero measure in $\mathbb{C}$. □

The incorporation of explicitly learnable damping enables D-LinOSS to model a wider range of stable dynamical systems, increasing the representational capacity of the overall deep sequence model. The empirical performance benefits are discussed in the following sections.

## 3.2 MOTIVATION

A natural question is whether or not a larger set of reachable eigenvalues is empirically useful. Notably, LinOSS is provably universal (Rusch & Rus, 2025; Lanthaler et al., 2024), meaning it can approximate any causal and continuous operator between input and output signals to arbitrary accuracy (see Appendix A.5 for a formal definition). This property trivially extends to D-LinOSS, as setting $\mathbf{G} = \mathbf{0}$ recovers LinOSS-IMEX. However, while universality characterizes theoretical capacity, it is not necessarily indicative of how well a model learns in practice. To motivate the empirical benefits of a broader spectral range, we study the following synthetic experiment.

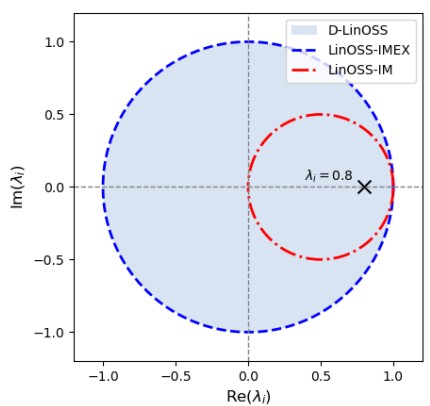

Figure 2: Reachable eigenvalue sets for the different oscillatory SSMs. The eigenvalue $\lambda = 0.8$ of the exponential decay experiment only lies in the spectral range of D-LinOSS.

We simulate a dynamical system with a single discrete eigenvalue $\lambda = 0.8$, corresponding to an exponentially decaying response. No input or output transformations are applied. Random sequences of scalar values are passed through this system, and models are trained to predict the resulting output. A small hyperparameter grid was searched for each model, three random seeds were trained per

Table 1: Learning exponential decay.

| Model | RMSE $\times 10^{-3}$ | |
|---|---|---|
| LinOSS-IMEX | $24.5 \pm 2.3$ | *30.6* $\times$ |
| LinOSS-IM | $8.0 \pm 1.7$ | *10.0* $\times$ |
| D-LinOSS | $0.8 \pm 0.1$ | *1.0* $\times$ |

configuration, and models are compared in terms of the best average test root mean squared error (RMSE) across seeds. (Appendix B.1 provides more detail on the experimental setup.)

D-LinOSS achieves test RMSE approximately $10\times$ lower than LinOSS-IM and $30\times$ lower than LinOSS-IMEX. Reflecting on the eigenvalue ranges depicted in Appendix A.4, the target eigenvalue $\lambda = 0.8$ lies outside the reachable spectra of both baseline models. This gap in performance suggests that although the previous LinOSS models are universal, a limited spectral range can impair the models' ability to represent certain temporal relationships. By directly learning system damping $\mathbf{G}$, D-LinOSS moves beyond rigidly defined energy dissipation behavior and is capable of accurately capturing a wider range of dynamics.

### 3.3 PARAMETERIZATION

During training, the learnable parameters $\mathbf{A}, \mathbf{G}$, and $\Delta t$ must satisfy the constraints detailed in Proposition 3.2 to ensure system stability. Unconstrained (denoted by bar) continuous-time SSM matrices and integration time steps are reparameterized through the following activations to enforce these conditions.

$$\Delta t = \sigma(\bar{\Delta}t), \quad \mathbf{G} = \text{ReLU}(\bar{\mathbf{G}}), \quad \mathbf{A} = \text{Clamp}(\bar{\mathbf{A}}, L(\mathbf{G}, \Delta t), U(\mathbf{G}, \Delta t))$$

Here, $\text{Clamp}(\cdot)$ bounds $\bar{\mathbf{A}}$ between lower bound $L$ and upper bound $U$, which are expressions derived from the quadratic inequality of the stability criterion $\mathcal{S}$.

### 3.4 INITIALIZATION

Performance of SSMs can be heavily impacted by the initialized distribution of eigenvalues (Orvieto et al., 2023). Many approaches, e.g., Gu et al. (2021); Smith et al. (2023), leverage structured initialization schemes such as the HiPPO framework (Gu et al. (2020)) to enhance long-range learning capability, but Orvieto et al. (2023) indicates that simpler initialization techniques, such as uniform ring initialization, are capable of recovering equal performance.

Given the parameterization in Section 3.3 and the bijective mapping between the parameter space $(\mathbf{A}_i, \mathbf{G}_i)$ and the eigenspace $(\lambda_i, \lambda_i^*)$ established in Proposition 3.3, we investigate initialization strategies for SSM layers based on uniform sampling in each of these spaces. Appendix B.6 details a comparison between two approaches: (i) uniformly sampling parameters $(\mathbf{A}_i, \mathbf{G}_i)$ across different ranges $[A_{\min}, A_{\max}]$ and $[G_{\min}, G_{\max}]$, and (ii) uniformly sampling eigenvalues within a complex ring across different radial bounds $[r_{\min}, r_{\max}]$ and angular bounds $[\theta_{\min}, \theta_{\max}]$. In the latter case, parameters are then obtained via the inverse mapping $(\mathbf{A}_i, \mathbf{G}_i) = \Phi^{-1}(\lambda_i)$ for all $i \in \{1, \ldots, m\}$. Our study indicates that sampling eigenvalues within the radial band $[0.9, 1.0]$ and the full angular range $[0, 2\pi)$ yields strong performance. This technique is used for all subsequent D-LinOSS results.

## 4 RESULTS

Following the experimental design from Rusch & Rus (2025); Walker et al. (2024); Zhou et al. (2021), we evaluate the empirical performance of D-LinOSS on a suite of real-world learning tasks that span disciplines across biology, medicine, chemistry, photonics, and climate. As the linear complexity and fixed state size of SSMs emphasize their utility for learning long-range dependencies, we evaluate candidate models on datasets with temporal relationships spanning thousands of measurements. We compare model performance with a total of sixteen other state-of-the-art sequence modeling approaches, including the precursor models LinOSS-IM and LinOSS-IMEX. Experimental design and hyperparameter spreads are kept consistent across all models to ensure fair comparison.

Table 2: Test accuracies averaged over five different seeds on UEA time-series classification datasets. The highest score is indicated in **bold** and the second highest is underlined. The dataset names are abbreviations of the following UEA datasets: EigenWorms (Worms), SelfRegulationSCP1 (SCP1), SelfRegulationSCP2 (SCP2), EthanolConcentration (Ethanol), Heartbeat, and MotorImagery (Motor).

| | **Worms** | **SCP1** | **SCP2** | **Ethanol** | **Heartbeat** | **Motor** | **Avg** |
|---|---|---|---|---|---|---|---|
| Seq. length | 17,984 | 896 | 1,152 | 1,751 | 405 | 3,000 | |
| # of Classes | 5 | 2 | 2 | 4 | 2 | 2 | |
| NRDE | $83.9 \pm 7.3$ | $80.9 \pm 2.5$ | $53.7 \pm 6.9$ | $25.3 \pm 1.8$ | $72.9 \pm 4.8$ | $47.0 \pm 5.7$ | 60.6 |
| NCDE | $75.0 \pm 3.9$ | $79.8 \pm 5.6$ | $53.0 \pm 2.8$ | $\underline{29.9 \pm 6.5}$ | $73.9 \pm 2.6$ | $49.5 \pm 2.8$ | 60.2 |
| Log-NCDE | $85.6 \pm 5.1$ | $83.1 \pm 2.8$ | $53.7 \pm 4.1$ | $\mathbf{34.4 \pm 6.4}$ | $75.2 \pm 4.6$ | $53.7 \pm 5.3$ | 64.3 |
| LRU | $87.8 \pm 2.8$ | $82.6 \pm 3.4$ | $51.2 \pm 3.6$ | $21.5 \pm 2.1$ | $\mathbf{78.4 \pm 6.7}$ | $48.4 \pm 5.0$ | 61.7 |
| S5 | $81.1 \pm 3.7$ | $\mathbf{89.9 \pm 4.6}$ | $50.5 \pm 2.6$ | $24.1 \pm 4.3$ | $\underline{77.7 \pm 5.5}$ | $47.7 \pm 5.5$ | 61.8 |
| S6 | $85.0 \pm 16.1$ | $82.8 \pm 2.7$ | $49.9 \pm 9.4$ | $26.4 \pm 6.4$ | $76.5 \pm 8.3$ | $51.3 \pm 4.7$ | 62.0 |
| Mamba | $70.9 \pm 15.8$ | $80.7 \pm 1.4$ | $48.2 \pm 3.9$ | $27.9 \pm 4.5$ | $76.2 \pm 3.8$ | $47.7 \pm 4.5$ | 58.6 |
| LinOSS-IMEX | $80.0 \pm 2.7$ | $87.5 \pm 4.0$ | $\mathbf{58.9 \pm 8.1}$ | $\underline{29.9 \pm 1.0}$ | $75.5 \pm 4.3$ | $57.9 \pm 5.3$ | 65.0 |
| LinOSS-IM | $\mathbf{95.0 \pm 4.4}$ | $87.8 \pm 2.6$ | $58.2 \pm 6.9$ | $\underline{29.9 \pm 0.6}$ | $75.8 \pm 3.7$ | $\underline{60.0 \pm 7.5}$ | $\underline{67.8}$ |
| D-LinOSS | $\underline{93.9 \pm 3.2}$ | $88.9 \pm 3.0$ | $\underline{58.6 \pm 2.3}$ | $\underline{29.9 \pm 0.6}$ | $75.8 \pm 4.9$ | $\mathbf{61.1 \pm 2.0}$ | $\mathbf{68.0}$ |

## 4.1 UEA TIME-SERIES CLASSIFICATION

We consider a benchmark from the University of East Anglia (UEA) Multivariate Time Series Classification Archive (UEA-MTSCA) Bagnall et al. (2018) introduced in Walker et al. (2024). This benchmark consists of six datasets chosen to evaluate the ability of sequence models to capture long-range interactions. The UEA datasets are classification tasks, ranging from classifying organisms from motion readings (*EigenWorms*) to classifying fluid alcohol percentage based on measurements of transmissive light spectra (*EthanolConcentration*). We precisely follow the experimental design proposed in Walker et al. (2024), conducting a model hyperparameter search over a grid of 162 predetermined configurations for each dataset. Further, each model instance is trained on five seeds, and the average test accuracy for the top performing model instances are reported. The high scoring hyperparameter configurations of D-LinOSS model instances are tabulated in Appendix B.4. All models use the same 70-15-15 train-validation-test data splits, controlled by the seed for a given trial. Model scores for LinOSS-IM and LinOSS-IMEX are sourced from Rusch & Rus (2025) and all other model scores are sourced from Walker et al. (2024).

Out of all models tested, D-LinOSS achieves the highest average test accuracy across the six UEA datasets–raising the previous high score from 67.8% to 68.0%. Notably, D-LinOSS improves state-of-the-art accuracy on MotorImagery by $1.1\%$ and scores in the top two for five out of the six datasets. D-LinOSS also outperforms the combination of both preceding models: the average score-wise maximum between LinOSS-IM and LinOSS-IMEX is $67.9\%$, still shy of D-LinOSS. D-LinOSS improves on or matches the second best model, LinOSS-IM, in all but one dataset, EigenWorms, which is the smallest dataset out of the six.

## 4.2 PPG-DALIA TIME-SERIES REGRESSION

We evaluate model performance on the *PPG dataset for motion compensation and heart rate estimation in Daily Life Activities* (PPG-DaLiA) (Reiss et al., 2019), which is a time-series regression task. Here, models are challenged with learning human heart rate patterns as a function of various sensor measurements, such as ECG readings, wearable accelerometers, and respiration sensing. The dataset consists of 15 different subjects performing a variety of daily tasks, and bio-sensory data is collected in sequences up to 50,000 points in length. We follow the same experimental design as before, searching model hyperparameters over a grid of 162 configurations and training each model instance on five seeds. All models use the same 70-15-15 data split. D-LinOSS achieves the best results, reducing the lowest MSE from 6.4 to 6.16 ($\times 10^{-2}$) compared to LinOSS-IM.

Table 3: Test accuracies averaged over five different seeds on the PPG-DaLiA time-series regression dataset. The best score is indicated in **bold** and the second best is underlined.

| Model | MSE $\times 10^{-2}$ |
|---|---|
| NRDE Morrill et al. (2021) | $9.90 \pm 0.97$ |
| NCDE (Kidger et al., 2020) | $13.54 \pm 0.69$ |
| Log-NCDE Walker et al. (2024) | $9.56 \pm 0.59$ |
| LRU Orvieto et al. (2023) | $12.17 \pm 0.49$ |
| S5 Smith et al. (2023) | $12.63 \pm 1.25$ |
| S6 Gu & Dao (2023) | $12.88 \pm 2.05$ |
| Mamba Gu & Dao (2023) | $10.65 \pm 2.20$ |
| LinOSS-IMEX Rusch & Rus (2025) | $7.5 \pm 0.46$ |
| LinOSS-IM Rusch & Rus (2025) | $\underline{6.4 \pm 0.23}$ |
| D-LinOSS | $\mathbf{6.16 \pm 0.73}$ |

Table 4: Mean absolute error on the weather dataset predicting the future 720 time steps based on the past 720 time steps. The best score is indicated in **bold** and the second best is underlined.

| Model | Mean Absolute Error |
|---|---|
| Informer (Zhou et al., 2021) | 0.731 |
| Informer[†] (Zhou et al., 2021) | 0.741 |
| LogTrans (Li et al., 2019) | 0.773 |
| Reformer (Kitaev et al., 2020) | 1.575 |
| LSTMa (Bahdanau et al., 2016) | 1.109 |
| LSTnet (Lai et al., 2018) | 0.757 |
| S4 (Gu et al., 2021) | 0.578 |
| LinOSS-IMEX Rusch & Rus (2025) | $\underline{0.508}$ |
| LinOSS-IM Rusch & Rus (2025) | 0.528 |
| D-LinOSS | **0.486** |

## 4.3 WEATHER TIME-SERIES FORECASTING

To assess the generality of D-LinOSS as a sequence-to-sequence model, as in Gu et al. (2021), we evaluate its performance on a long-horizon time-series forecasting task without any architectural modifications. In this setting, forecasting is framed as a masked sequence transformation problem, allowing the model to predict future values based solely on partially masked input sequences.

We focus on the weather forecasting task introduced in Zhou et al. (2021), which involves predicting one month of hourly measured multivariate local climatological data based on the previous month's measurements. The dataset spans 1,600 U.S. locations between 2010 to 2013; further details are provided in NCEI (2025).

In this benchmark, D-LinOSS is compared against Transformer-based architectures, LSTM variants, the structured state-space model S4, and previous versions of LinOSS. D-LinOSS achieves the best performance, reducing the lowest mean absolute error (MAE) from 0.508 (LinOSS-IMEX) to 0.486.

## 4.4 SYNTHETIC ADDING

An additional practical benefit of D-LinOSS is faster training convergence compared to preceding models. To showcase this, we consider the synthetic adding task (Hochreiter, 1997), where the model must compute the sum of two randomly specified numbers embedded in a sequence of white noise. To make the task challenging, models are required to predict the sum using only the final output token, rather than aggregating all outputs across the sequence. We evaluate sequence lengths of 500,

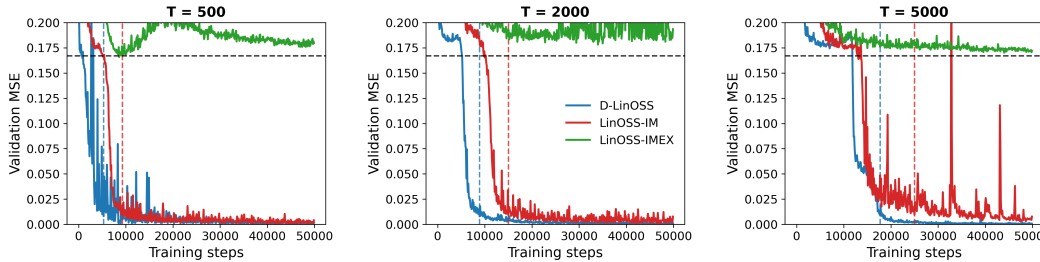

Figure 3: Validation metric convergence for the adding task of different sequence lengths.

2000, and 5000, and perform a small grid search for all LinOSS variants. Further details are provided in Appendix B.3.

Comparing the best-performing instances, we find that D-LinOSS is not only more capable of reaching a correct solution but it also converges faster. As shown in Figure 3, D-LinOSS attains a validation MSE of $10^{-2}$ about $1.7\times$ faster than LinOSS-IM for both sequence lengths of 500 and 2000 and about $1.4\times$ faster for length 5000. In contrast, LinOSS-IMEX fails to outperform the baseline of random guessing (with MSE of 0.167) for any task. These results highlight the greater flexibility and expressivity of D-LinOSS, which translate into more efficient and consistent learning.

## 5 RELATED WORK

State Space Models (SSMs) were introduced as a deep learning framework for sequential data in Gu et al. (2021). Early variants (Gu et al., 2022; Nguyen et al., 2022; Goel et al., 2022) relied on Fast Fourier Transform (FFT) and HiPPO parameterizations (Gu et al., 2020) to efficiently compute linear recurrences. More recent architectures employ diagonal state matrices in combination with fast associative parallel scans, which was first developed for RNNs (Martin & Cundy, 2017; Kaul, 2020) and later adapted to SSMs in Smith et al. (2023). While these models initially required HiPPO matrices to initialize weights, subsequent work has shown that simple random initialization is sufficient (Orvieto et al., 2023). Finally, D-LinOSS and the models above are based on LTI systems, there is growing interest in time-varying SSMs for challenging domains such as language and video (Gu & Dao, 2023; Dao & Gu, 2024; Hasani et al., 2022; Merrill et al., 2024).

The most closely related model to our proposed D-LinOSS is the original LinOSS, introduced in Rusch & Rus (2025). While LinOSS was the first SSM to explicitly leverage oscillatory dynamics, the idea of incorporating oscillatory behavior into deep learning architectures has also appeared in other domains. For instance, recurrent models such as coupled oscillatory RNNs (coRNNs) (Rusch & Mishra, 2021a) and UnICORNNs (Rusch & Mishra, 2021b) introduce oscillations into their hidden state dynamics, while graph-based approaches like Graph Coupled Oscillator Networks (GraphCON) (Rusch et al., 2022) extend similar principles to structured data.

## 6 DISCUSSION AND CONCLUSION

We introduced D-LinOSS, an extension of the LinOSS model that incorporates learnable damping across arbitrary time scales. Through spectral analysis, we showed that existing LinOSS variants are rigidly defined by their discretization scheme and can only express a narrow set of dynamical behaviors. In contrast, D-LinOSS captures the full range of stable second-order dynamics.

This expanded expressivity yields a $10-30\times$ improvement on a synthetic regression task, leads to consistent performance gains across eight real-world benchmarks, and enables faster convergence on the adding task. D-LinOSS outperforms all baselines considered in this work, including Transformer-based models, LSTM variants, other modern SSMs, as well as previous versions of LinOSS. Additionally, D-LinOSS reduces the LinOSS hyperparameter search space without adding any computational overhead. These results establish D-LinOSS as an efficient and powerful extension to the family of deep state space models.

While D-LinOSS demonstrates strong empirical results as a general sequence model, it is based on layers of LTI systems, which are fundamentally limited in their ability to capture certain contextual dependencies, such as selective copying (Gu & Dao (2023), Jing et al. (2019)). Building on the growing interest in time-varying SSMs sparked by Gu & Dao (2023); Dao & Gu (2024), we aim to explore future work on variants of D-LinOSS that integrate the selectivity of time-varying dynamics.

As D-LinOSS is inherently well-suited to represent temporal relationships with oscillatory structure, we aim to explore applications where such patterns are fundamental. In particular, climate science, seismic monitoring, and astrophysics data all exhibit complex patterns governed by oscillatory behavior. Moving forward, we believe that D-LinOSS will be a key player in advancing machine-learning based approaches in domains grounded in the physical sciences.

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

**Supplementary Material for:**
Learning to Dissipate Energy in Oscillatory State-Space Models



## A THEORETICAL PROPERTIES

### A.1 D-LinOSS EIGENVALUES

**Proposition A.1.** *The eigenvalues of the D-LinOSS recurrent matrix* $\mathbf{M} \in \mathbb{R}^{2m \times 2m}$ *are*

$$\lambda_{i_{1,2}} = \frac{1 + \frac{\Delta t_i}{2}\mathbf{G}_i - \frac{\Delta t_i^2}{2}\mathbf{A}_i}{1 + \Delta t_i \mathbf{G}_i} \pm \frac{\frac{\Delta t_i}{2}\sqrt{(\mathbf{G}_i - \Delta t_i \mathbf{A}_i)^2 - 4\mathbf{A}_i}}{1 + \Delta t_i \mathbf{G}_i},$$

*where pairs of eigenvalues are denoted as* $\lambda_{i_{1,2}}$ *and* $i = 1, 2, ..., m$.

*Proof.* $\mathbf{M}$ (6) is a matrix with diagonal sub-blocks $\mathbf{M}_{11}$, $\mathbf{M}_{12}$, $\mathbf{M}_{21}$, and $\mathbf{M}_{22}$, i.e. it follows the structure:

$$\mathbf{M} = \begin{Vmatrix} \begin{bmatrix} (\mathbf{M}_{11})_1 & & \\ & \ddots & \\ & & (\mathbf{M}_{11})_m \end{bmatrix} & \begin{bmatrix} (\mathbf{M}_{12})_1 & & \\ & \ddots & \\ & & (\mathbf{M}_{12})_m \end{bmatrix} \\ \begin{bmatrix} (\mathbf{M}_{21})_1 & & \\ & \ddots & \\ & & (\mathbf{M}_{21})_m \end{bmatrix} & \begin{bmatrix} (\mathbf{M}_{22})_1 & & \\ & \ddots & \\ & & (\mathbf{M}_{22})_m \end{bmatrix} \end{Vmatrix}$$

Since $\mathbf{M}$ represents $m$ uncoupled oscillatory systems, we see that the operation $\mathbf{Mw}$, $\mathbf{w}^{\mathsf{T}} = [\mathbf{z}^{\mathsf{T}}, \mathbf{x}^{\mathsf{T}}]$ can be independently split across the $m$ pairs of state variables $[\mathbf{z}_i, \mathbf{x}_i]$. Thus, it suffices to re-arrange $\mathbf{M}$ into $m$ different $2 \times 2$ systems and analyze the eigenvalues of each system separately. Substituting the real expressions for $\mathbf{M}_{11}$, $\mathbf{M}_{12}$, $\mathbf{M}_{21}$, and $\mathbf{M}_{22}$, and using brackets to denote the $2 \times 2$ matrix formed by diagonally indexing along the sub-blocks of $\mathbf{M}$, the $i^{th}$ system matrix is:

$$[\mathbf{M}]_i = \begin{bmatrix} \mathbf{S}_i^{-1} & -\Delta t_i \mathbf{S}_i^{-1}\mathbf{A}_i \\ \Delta t_i \mathbf{S}_i^{-1} & 1 - \Delta t_i^2 \mathbf{S}_i^{-1}\mathbf{A}_i \end{bmatrix}$$

A straightforward exercise using the definition $\mathbf{S} = \mathbf{I} + \Delta t \mathbf{G}$ leads us to the $i^{th}$ pair of eigenvalues.

$$\det(\lambda \mathbf{I} - [\mathbf{M}]_i) = \det \begin{bmatrix} \lambda - \mathbf{S}_i^{-1} & \Delta t_i \mathbf{S}_i^{-1}\mathbf{A}_i \\ -\Delta t_i \mathbf{S}_i^{-1} & \lambda - 1 + \Delta t_i^2 \mathbf{S}_i^{-1}\mathbf{A}_i \end{bmatrix}$$

$$= (\lambda - \mathbf{S}_i^{-1})(\lambda - 1 + \Delta t_i^2 \mathbf{S}_i^{-1}\mathbf{A}_i) + (\Delta t_i \mathbf{S}_i^{-1})(\Delta t_i \mathbf{S}_i^{-1}\mathbf{A}_i)$$

$$= \lambda^2 + \lambda(\mathbf{S}_i^{-1}(\Delta t_i^2 \mathbf{A}_i - 1) - 1) + \mathbf{S}_i^{-1}$$

$$= \lambda^2 + \lambda \frac{-2 - \Delta t_i \mathbf{G}_i + \Delta t_i^2 \mathbf{A}_i}{1 + \Delta t_i \mathbf{G}_i} + \frac{1}{1 + \Delta t_i \mathbf{G}_i} = 0$$

$$\implies \lambda_{i_{1,2}} = \frac{1 + \frac{\Delta t_i}{2}\mathbf{G}_i - \frac{\Delta t_i^2}{2}\mathbf{A}_i}{1 + \Delta t_i \mathbf{G}_i} \pm \frac{\frac{\Delta t_i}{2}\sqrt{(\mathbf{G}_i - \Delta t_i \mathbf{A}_i)^2 - 4\mathbf{A}_i}}{1 + \Delta t_i \mathbf{G}_i}$$

$$\square$$

## A.2 STABILITY CRITERION

**Proposition A.2.** *Assume $\mathbf{A}_i$, $\mathbf{G}_i$ are non-negative, and $\Delta t_i \in (0, 1]$. If the following is satisfied:*

$$(\mathbf{G}_i - \Delta t_i \mathbf{A}_i)^2 \leq 4\mathbf{A}_i \tag{11}$$

*Then $\lambda_{i_{1,2}}$ come in complex conjugate pairs $\lambda_i$, $\lambda_i^*$ with the following magnitude:*

$$|\lambda_i| = \frac{1}{\sqrt{1 + \Delta t_i \mathbf{G}_i}} \leq 1,$$

*i.e. the eigenvalues are unit-bounded. Define $\mathcal{S}_i$ to be the set of all $(\mathbf{A}_i, \mathbf{G}_i)$ that satisfy the above condition. For notational convenience, we order the eigenvalues such that $\mathrm{Im}(\lambda_i) \geq 0$, $\mathrm{Im}(\lambda_i^*) \leq 0$.*

*Proof.* $(\mathbf{G}_i - \Delta t_i \mathbf{A}_i)^2 - 4\mathbf{A}_i$ is the determinant of each eigenvalue pair, so $(\mathbf{G}_i - \Delta t_i \mathbf{A}_i)^2 \leq 4\mathbf{A}_i$ means $\lambda_i$ can be written in complex form with the following real and imaginary components.

$$\mathrm{Re}(\lambda_i) = \frac{1 + \frac{\Delta t_i}{2}\mathbf{G}_i - \frac{\Delta t_i^2}{2}\mathbf{A}_i}{1 + \Delta t_i \mathbf{G}_i}$$

$$\mathrm{Im}(\lambda_i) = \pm \frac{\frac{\Delta t_i}{2}\sqrt{4\mathbf{A}_i - (\mathbf{G}_i - \Delta t_i \mathbf{A}_i)^2}}{1 + \Delta t_i \mathbf{G}_i}$$

The magnitude of this complex number is:

$$|\lambda_i| = \sqrt{\mathrm{Re}(\lambda_i)^2 + \mathrm{Im}(\lambda_i)^2} = \frac{1}{\sqrt{1 + \Delta t_i \mathbf{G}_i}}$$

$$\Delta t_i, \mathbf{G}_i \geq 0 \implies |\lambda_i| \leq 1$$

We note that this stability criterion is a sufficient but not necessary condition. There exists solutions $(\mathbf{A}_i, \mathbf{G}_i)$ rendering $|\lambda_i| \leq 1$ that do not lie in $\mathcal{S}_i$. However, as shown in Proposition 3.3, there already exists a bijection from $\mathcal{S}_i$ to the full complex unit disk.

$\square$

## A.3 SPECTRAL IMAGE OF D-LINOSS

**Proposition A.3.** *The mapping $\Phi : \mathcal{S}_i \to \mathbb{C}_{|z| \leq 1} \setminus \{0\}$ defined by $(\mathbf{A}_i, \mathbf{G}_i) \mapsto (\lambda_i, \lambda_i^*)$ is bijective.*

*Proof.* The "bijective relationship between $(\mathbf{A}_i, \mathbf{G}_i) \in \mathcal{S}_i$ and $(\lambda_i, \lambda_i^*) \in \mathbb{C}_{|z| \leq 1} \setminus \{0\}$" is an abuse of notation for the bijective relationship between $(\mathbf{A}_i, \mathbf{G}_i) \in \mathcal{S}_i$ and just the first eigenvalue $\lambda_{i_1}$ (with non-negative imaginary part) over the half-disk $\mathbb{C}_{|z| \leq 1, \mathrm{Im}(z) \geq 0} \setminus \{0\}$. Conjugate symmetry of the eigenvalues then "fills out" the full space $\mathbb{C}_{|z| \leq 1} \setminus \{0\}$.

As such, we aim to show $\Phi_1 : \mathcal{S}_i \to \mathbb{C}_{|z| \leq 1, \mathrm{Im}(z) \geq 0} \setminus \{0\}, (\mathbf{A}_i, \mathbf{G}_i) \mapsto \lambda_i$ is a bijection. We first show that $\Phi_1$ is bijective over some larger region; equivalently, $\Phi_1$ has a well-defined inverse function $\Psi_1$ (a function $\Psi_1$ such that $\Phi_1 \circ \Psi_1 = \mathrm{id}_Y$ and $\Psi_1 \circ \Phi_1 = \mathrm{id}_X$). The first relation is equivalent to surjectivity: $\Phi_1(\Psi_1(y)) = y$ means all $y$ are reachable through $\Phi_1$ via $x = \Psi_1(y)$. The second relation is equivalent to injectivity: $\Psi_1(\Phi_1(x)) = x$ means all $x$ and $y$ such that $\Phi_1(x) = \Phi_1(y)$ must satisfy $x = y$. Afterward, we show that the image of $\Phi_1$ is correct, i.e. $\Phi_1(\mathcal{S}_i) = \mathbb{C}_{|z| \leq 1, \mathrm{Im}(z) \geq 0} \setminus \{0\}$, which shows that $\Phi_1$ is bijective with respect to the desired regions.

We assume constant $\Delta t_i \in (0, 1]$. Consider the function $\Psi_1 : \lambda_i \mapsto (\mathbf{A}_i, \mathbf{G}_i)$ as defined below.

$$\mathbf{A}_i = \frac{\lambda_i \lambda_i^* - \lambda_i - \lambda_i^* + 1}{\Delta t_i^2 \lambda_i \lambda_i^*}, \quad \mathbf{G}_i = \frac{1 - \lambda_i \lambda_i^*}{\Delta t_i \lambda_i \lambda_i^*}$$

Plugging the expressions for $\mathbf{A}_i$, $\mathbf{G}_i$ above into $\Phi_1$ and doing some algebra reveals the intermediate result:

$$\mathrm{Re}(\Phi_1(\Psi_1(\lambda_i))) = \left.\frac{1 + \frac{\Delta t_i}{2}\mathbf{G}_i - \frac{\Delta t_i^2}{2}\mathbf{A}_i}{1 + \Delta t_i \mathbf{G}_i}\right|_{(\mathbf{A}_i, \mathbf{G}_i) = \Psi_1(\lambda_i)} = \frac{1}{2}(\lambda_i + \lambda_i^*)$$

$$\mathrm{Im}(\Phi_1(\Psi_1(\lambda_i))) = \left.\frac{\frac{\Delta t_i}{2}\sqrt{(\mathbf{G}_i - \Delta t_i\mathbf{A}_i)^2 - 4\mathbf{A}_i}}{1 + \Delta t_i \mathbf{G}_i}\right|_{(\mathbf{A}_i, \mathbf{G}_i) = \Psi_1(\lambda_i)} = \frac{1}{2}(\lambda_i - \lambda_i^*)$$

$$\Rightarrow \Phi_1(\Psi_1(\lambda_i)) = \lambda_i$$

Similarly, plugging the eigenvalue expression for $\lambda_i$ into $\Psi_1$ and doing some algebra shows:

$$\Psi_1(\Phi_1((\mathbf{A}_i, \mathbf{G}_i))_0 = \left.\frac{\lambda_i\lambda_i^* - \lambda_i - \lambda_i^* + 1}{\Delta t_i^2 \lambda_i \lambda_i^*}\right|_{\lambda_i = \Phi_1((\mathbf{A}_i, \mathbf{G}_i))} = \mathbf{A}_i$$

$$\Psi_1(\Phi_1((\mathbf{A}_i, \mathbf{G}_i))_1 = \left.\frac{1 - \lambda_i\lambda_i^*}{\Delta t_i \lambda_i \lambda_i^*}\right|_{\lambda_i = \Phi_1((\mathbf{A}_i, \mathbf{G}_i))} = \mathbf{G}_i$$

$$\Rightarrow \Psi_1(\Phi_1((\mathbf{A}_i, \mathbf{G}_i)) = (\mathbf{A}_i, \mathbf{G}_i)$$

So $\Phi_1$, $\Psi_1$ are well-defined inverses of each other. In the above derivations, we used the fact that $\lambda_i \neq 0$.

It remains to show that $\Phi_1(\mathcal{S}_i) = \mathbb{C}_{|z| \leq 1,\, \mathrm{Im}(z) \geq 0} \setminus \{0\}$. Proposition 3.2 has already shown that $\Phi(\mathcal{S}_i) \subseteq \mathbb{C}_{|z| \leq 1,\, \mathrm{Im}(z) \geq 0} \setminus \{0\}$, so it remains to show the reverse set inclusion $\Psi_1(\mathbb{C}_{|z| \leq 1,\, \mathrm{Im}(z) \geq 0} \setminus \{0\}) \subseteq \mathcal{S}_i$.

It's clear that $\mathbf{A}_i$, $\mathbf{G}_i$ are non-negative as the numerators are denominators of the expressions in $\Psi_1$ are both non-negative given $0 < |\lambda_i| \leq 1$. Now freely leveraging the inverse function, we see the final stability inequality is also satisfied:

$$(\mathbf{G}_i - \Delta t_i\mathbf{A}_i)^2 - 4\mathbf{A}_i = \frac{(\lambda_i - \lambda_i^*)^2}{(\Delta t_i \lambda_i \lambda_i^*)^2} = \frac{-4\mathrm{Im}(\lambda_i)^2}{(\Delta t_i \lambda_i \lambda_i^*)^2} \leq 0$$

In other words, parameters $(\mathbf{A}_i, \mathbf{G}_i)$ derived from the inverse map $\Psi(\lambda_i)$ lie within the stable region $\mathcal{S}_i$ when $\lambda_i \in \mathbb{C}_{|z| \leq 1,\, \mathrm{Im}(z) \geq 0} \setminus \{0\}$, which is exactly the reverse set inclusion.

$\square$

### A.4 SET MEASURE OF LINOSS EIGENVALUES

**Proposition A.4.** *For both LinOSS-IM and LinOSS-IMEX, the set of eigenvalues constructed from* $\mathbf{A}_i \in \mathbb{R}_{\geq 0}$ *and* $\Delta t_i \in (0, 1]$ *is of measure zero in* $\mathbb{C}$.

*Proof.* Recall the eigenvalue expressions from Rusch & Rus (2025):

$$\lambda_{i_{1,2}}^{\mathrm{IM}} = \frac{1}{1 + \Delta t_i^2 \mathbf{A}_i} \pm j\frac{\Delta t_i\sqrt{\mathbf{A}_i}}{1 + \Delta t_i^2 \mathbf{A}_i}, \quad \lambda_{i_{1,2}}^{\mathrm{IMEX}} = \frac{1}{2}(2 - \Delta t_i^2 \mathbf{A}_i) \pm \frac{j}{2}\sqrt{\Delta t_i^2 \mathbf{A}_i(4 - \Delta t_i^2 \mathbf{A}_i)},$$

Since $\lambda_{i_1} = \lambda_{i_2}^*$, it suffices to prove the proposition for the first eigenvalue, i.e. $\{\lambda_{i_1} \in \mathbb{C} \mid \mathbf{A}_i \in \mathbb{R}_{\geq 0}, \Delta t_i \in (0, 1]\}$ is a set of measure zero. We start with the following lemma:

**Lemma A.5.** *Let $f : M \to N$ be a continuously differentiable map of manifolds where $dim M < dim N$. Then $f(M)$ is of measure zero in N (Gualtieri, 2016).*

Using Lemma A.5, it suffices to show that range($\lambda_{i_1}^{\text{IM}}$) is a 1-manifold in $\mathbb{C} \cong \mathbb{R}^2$. Beginning with the LinOSS-IM expression, apply the change of variables $\gamma_i := \Delta t_i \sqrt{\mathbf{A}_i}$, where $\mathbf{A}_i \in \mathbb{R}_{\geq 0}, \Delta t_i \in (0, 1] \iff \gamma_i \in \mathbb{R}_{\geq 0}$:

$$\lambda_{i_1}^{\text{IM}}(\gamma_i) = \left( \frac{1}{1 + \gamma_i^2}, \frac{\gamma_i}{1 + \gamma_i^2} \right)$$

This map is continuously differentiable, injective on $\mathbb{C}$ (and surjective onto its image), and its inverse is continuously differentiable, so it satisfies the conditions of Lemma A.5. So, the range of $\lambda_{i_1}^{\text{IM}}$ is an embedded 1-manifold in $\mathbb{R}^2$ and therefore a set of measure zero.

Using the same change of variables, an identical argument can be applied to the LinOSS-IMEX expression to show it also produces a set of measure zero; this argument is omitted for concision.

$\square$

### A.5 UNIVERSALITY

An operator $\Phi : C_0([0, T]; \mathbb{R}^p) \to C_0([0, T]; \mathbb{R}^q)$ is said to be *causal* if $\forall t \in [0, T]$, if $\mathbf{u}, \mathbf{v} \in C_0([0, T]; \mathbb{R}^p)$ are two input signals such that $\mathbf{u}|_{[0,t]} \equiv \mathbf{v}|_{[0,t]}$, then $\Phi(\mathbf{u})(t) = \Phi(\mathbf{v})(t)$.

An operator is said to be *continuous* if $\Phi : (C_0([0, T]; \mathbb{R}^p), \|\cdot\|_\infty) \to (C_0([0, T]; \mathbb{R}^q), \|\cdot\|_\infty)$, i.e. $\Phi$ is a map between continuous functions with respect to the $L^\infty$-norms on the input/output signals.

The theorem of Rusch & Rus (2025) is briefly paraphrased:

**Theorem A.6.** *Let $\Phi$ be any causal and continuous operator. Let $K \subset C_0([0, T]; \mathbb{R}^p)$ be compact. Then for any $\epsilon > 0$, there exists a configuration of hyperparameters and weight matrices, such that the output $\mathbf{z} : [0, T] \to \mathbb{R}^q$ of a LinOSS block satisfies:*

$$\sup_{t \in [0,T]} |\Phi(\mathbf{u})(t) - \mathbf{z}(t)| \leq \epsilon, \quad \forall \mathbf{u} \in K.$$

In other words, a LinOSS block can approximate any causal and continuous operator on compact input spaces to arbitrarily high accuracy.

## B EXPERIMENTS AND RESULTS

### B.1 REGRESSION EXPERIMENT

A small hyperparameter grid search was conducted over hidden dimension $\in \{8, 64\}$, SSM dimension $\in \{8, 64\}$, and number of blocks $\in \{2, 6\}$. Learning rate was kept constant at 1e-3.

Input sequences were sampled from white noise and passed through the discrete state-space system corresponding to parameters $A = 0.8, B = 1, C = 1, D = 0$. A sequence length of 1000 was selected to study the model behaviors within the regime of long range learning.

### B.2 VISUALIZING EIGENVALUES

The learned eigenvalue distributions across each layer are plotted for the highest-performing D-LinOSS, LinOSS-IM, and LinOSS-IMEX model instances on the MotorImagery classification task. Qualitatively, the learned eigenvalues are roughly distributed similarly to the initialized distribution. Recall that particular initialization techniques have been developed for each model that initializes eigenvalues only within a subset of the full range depicted in Figure 2. In the case of D-LinOSS, eigenvalues are initialized via uniform sampling in the magnitude band $|\lambda| \in [0.9, 1.0]$, whereas previous LinOSS uniformly sample the underlying $\mathbf{A}$ and $\Delta t$ parameters.

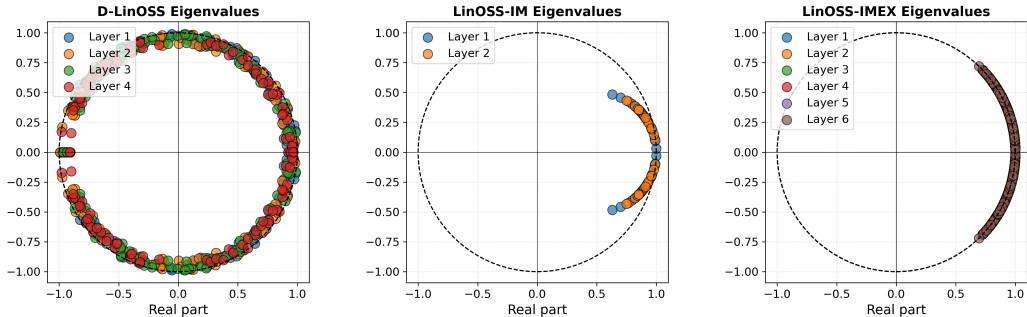

Figure 4: Learned eigenvalues across layers for best model instances on the **MotorImagery** task.

### B.3 ADDING TASK

Models were varied across number of blocks $\in \{2, 4, 6\}$ while keeping constant hidden dimension $= 128$, SSM dimension $= 128$, and learning rate $= 1e-4$. Each model configuration was trained on 5 different seeds and the highest performing model instances (as measured by fastest validation set convergence time) are compared in Figure 3.

### B.4 HYPERPARAMETERS

For the UEA-MTSCA classification and PPG regression experiments of Section 4.1 and Section 4.2, model hyperparameters were searched over a grid of 162 total configurations defined by Walker et al. (2024). This grid consists of learning rate $\in \{$1e-3, 1e-4, 1e-5$\}$, hidden dimension $\in \{16, 64, 128\}$, SSM dimension $\in \{16, 64, 256\}$, number of SSM blocks $\in \{2, 4, 6\}$, and inclusion of time $\in \{$True, False$\}$. For the weather forecasting experiment of Section 4.3, we instead perform random search over the same grid, except we sample learning rate continuously from a log-uniform distribution and allow for odd-numbered selections of the number of blocks.

Table 5: Best performing hyperparameters for D-LinOSS across each of the eight datasets.

| Dataset | lr | hidden dim | state dim | num blocks | include time |
|---|---|---|---|---|---|
| **Worms** | 1e-3 | 128 | 64 | 2 | False |
| **SCP1** | 1e-4 | 128 | 256 | 6 | True |
| **SCP2** | 1e-5 | 128 | 64 | 6 | False |
| **Ethanol** | 1e-5 | 16 | 256 | 4 | False |
| **Heartbeat** | 1e-4 | 16 | 16 | 2 | False |
| **Motor** | 1e-3 | 16 | 64 | 4 | False |
| **PPG** | 1e-3 | 64 | 64 | 4 | True |
| **Weather** | 7.95e-5 | 128 | 128 | 3 | False |

### B.5 COMPUTE REQUIREMENTS

Below, we tabulate the compute resources required for each model across all datasets considered in the UEA-MTSCA classification experiments of Section 4.1. The main table is sourced from Rusch & Rus (2025), which lists the total number of parameters, average GPU memory usage measured in MB, and average run time per 1000 training steps measured in seconds. All models operate on the same codebase and python libraries, adopted from both Walker et al. (2024) and Rusch & Rus (2025). These compute requirements are evaluated using an Nvidia RTX 4090 GPU.

Table 6: Compute requirements for the classification experiments considered in Section 4.1.

| | | NRDE | NCDE | Log-NCDE | LRU | S5 | Mamba | S6 | LinOSS-IMEX | LinOSS-IM | D-LinOSS |
|---|---|---|---|---|---|---|---|---|---|---|---|
| **Worms** | $|\theta|$ | 105110 | 166789 | 37977 | 101129 | 22007 | 27381 | 15045 | 26119 | 134279 | 134279 |
| | mem. | 2506 | 2484 | 2510 | 10716 | 6646 | 13486 | 7922 | 6556 | 3488 | 3488 |
| | time | 5386 | 24595 | 1956 | 94 | 31 | 122 | 68 | 37 | 14 | 14 |
| **SCP1** | $|\theta|$ | 117187 | 166274 | 91557 | 25892 | 226328 | 184194 | 24898 | 447944 | 991240 | 992776 |
| | mem. | 716 | 694 | 724 | 960 | 1798 | 1110 | 904 | 4768 | 4772 | 4790 |
| | time | 1014 | 973 | 635 | 9 | 17 | 7 | 3 | 42 | 38 | 38 |
| **SCP2** | $|\theta|$ | 200707 | 182914 | 36379 | 26020 | 5652 | 356290 | 26018 | 448072 | 399112 | 399496 |
| | mem. | 712 | 692 | 714 | 954 | 762 | 2460 | 1222 | 4772 | 2724 | 2736 |
| | time | 1404 | 1251 | 583 | 9 | 9 | 32 | 7 | 55 | 22 | 22 |
| **Ethanol** | $|\theta|$ | 93212 | 133252 | 31452 | 76522 | 76214 | 1032772 | 5780 | 70088 | 6728 | 71112 |
| | mem. | 712 | 692 | 710 | 1988 | 1520 | 4876 | 938 | 4766 | 1182 | 4774 |
| | time | 2256 | 2217 | 2056 | 16 | 9 | 255 | 4 | 48 | 8 | 37 |
| **Heartbeat** | $|\theta|$ | 15657742 | 1098114 | 168320 | 338820 | 158310 | 1034242 | 6674 | 29444 | 10936 | 4356 |
| | mem. | 6860 | 1462 | 2774 | 1466 | 1548 | 1650 | 606 | 922 | 928 | 672 |
| | time | 9539 | 1177 | 826 | 8 | 11 | 34 | 4 | 4 | 7 | 4 |
| **Motor** | $|\theta|$ | 1134395 | 186962 | 81391 | 107544 | 17496 | 228226 | 52802 | 106024 | 91844 | 20598 |
| | mem. | 4552 | 4534 | 4566 | 8646 | 4616 | 3120 | 4056 | 12708 | 4510 | 4518 |
| | time | 7616 | 3778 | 730 | 51 | 16 | 35 | 34 | 128 | 11 | 20 |

## B.6 INITIALIZATION TECHNIQUES

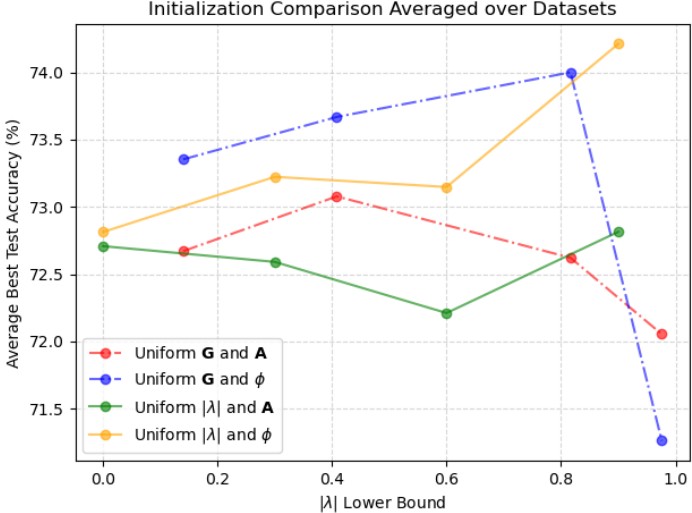

Figure 5: Initialization study varying intervals of eigenvalue magnitude and methods of sampling.

To understand how to best initialize the D-LinOSS parameters, we conduct a study on model performance comparing four different sampling techniques.

- Experiment 1: Uniformly sample $\mathbf{G} \in [0, G_{\max}]$ for different values of $G_{\max}$ and uniformly sample $\mathbf{A} \in [0, 1]$
- Experiment 2: Uniformly sample $\mathbf{G} \in [0, G_{\max}]$ for different values of $G_{\max}$ and uniformly sample $\phi \in [0, \pi)$
- Experiment 3: Radially uniform sample $|\lambda| \in [r_{\min}, 1]$ for different values of $r_{\min}$ and uniformly sample $\mathbf{A} \in [0, 1]$
- Experiment 4: Radially uniform sample $|\lambda| \in [r_{\min}, 1]$ for different values of $r_{\min}$ and uniformly sample $\phi \in [0, \pi)$

Where sampling $\mathbf{G}$ or sampling $|\lambda|$ are two different ways of controlling eigenvalue magnitude, and sampling $\mathbf{A}$ or sampling $\phi := \arg(\lambda)$ are two different ways of controlling eigenvalue phase. Note

that eigenvalues come in complex conjugate pairs, so sampling a single eigenvalue in $\phi \in [0, \pi)$ covers the full complex disk. "Radially uniform sampling" refers to uniformly sampling over the area of the cartesian coordinate ring described by the polar coordinate boundaries as opposed to uniformly sampling in the polar coordinates themselves.

For each experiment, values of $G_{\max}$ or $r_{\min}$ are varied to generate different eigenvalue distributions. For each evaluation of an initialization technique (a single point on figure 5), D-LinOSS is trained over a small sweep of 16 hyper-parameter configurations, and each model configuration is trained on 5 different seeds. This process is repeated for the three datasets SelfRegulationSCP1, Heartbeat, and MotorImagery. Figure 5 displays the average highest test score over all three datasets.

Radially uniform sampling eigenvalue magnitude and uniformly sampling eigenvalue phase is the best performing initialization technique observed. For these three datasets, a lower bound of $r_{\min} = 0.9$ results in the highest average test accuracy.

## C  ADDITIONAL EXPERIMENTS

### C.1  VARIATIONS OF ADDING

We develop additional variations of the Adding task to interpret how the underlying learnable damping mechanism relates to long-range learning. In particular, we conduct a parameter saliency ("sensitivity" or "importance") analysis to understand what subset of the learned recurrent matrix eigenvalues (modulated by the damping parameter $\mathbf{G}$) are most important for a given task. Further, we compare how different distributions of eigenvalues impact performance on task variations designed to express long-range dependencies.

As before, we consider the task of computing the sum of two numbers randomly selected within a sequence of uniform noise. However, we study two new task variations, both on sequences of length 1000: 1. the selected indices are always chosen within the first 100 elements, and 2. the selected indices are always chosen within the last 100 elements. As the model prediction is defined as the SSM's final sequential output, the first variation skews model requirements toward longer-range memory retention, whereas the second variation does not necessitate the same level of memory.

We begin by training D-LinOSS on these tasks, defaulting hyperparameters to hidden dimension = 128, SSM dimension = 128, learning rate = 1e-4, number of blocks = 3, and inclusion of time = False. Eigenvalue magnitude initialization range is set to $|\lambda| \in [0.5, 1.0]$ to capture a large spectral range for interpretation. 5 random seeds are trained and the best performing model instance is analyzed.

Given a trained model instance, the importance of each eigenvalue can be measured via the loss function's sensitivity to changes in the corresponding model parameters. We follow methods in classical neural network compression theory (LeCun et al., 1989; Hassibi & Stork, 1992) to measure parameter *saliency* using second-order derivative information. In particular, $L_i$, the saliency of the $i$'th weight $W_i$, is defined as the approximate increase in loss when $W_i$ is eliminated from the network.

$$L_i = \frac{1}{2} \frac{W_i^2}{(H^{-1})_{ii}} \tag{12}$$

$H = \nabla^2 l(W; X, y)$ is the Hessian matrix of the loss function taken with respect to model parameters $W$. Materializing and inverting the full matrix is computationally intractable for large networks, so we follow standard practice in using the simplifying assumption $(H^{-1})_{ii} \approx (H_{ii})^{-1}$. In the D-LinOSS parameterization, each eigenvalue $\lambda_i$ is a function of the corresponding parameters $\mathbf{A}_i, \mathbf{G}_i, \Delta t_i$, so we approximate the saliency of an eigenvalue $L_{\lambda_i} \approx L_{\mathbf{A}_i} + L_{\mathbf{G}_i} + L_{\Delta t_i}$.

Figure 6 shows eigenvalues with larger magnitude (equivalently, smaller $\mathbf{G}_i$), are more important to computing the sum in the First-100 Adding task. Figure 7 shows a similar trend but less pronounced, indicating that the low-damping recurrent modes are not as important to the Last-100 Adding task. This relationship is more precisely viewed in Figure 8; comparing the best-fit exponential functions $L_\lambda \approx c_1 e^{5.5|\lambda|}$ (First-100 Adding) to $L_\lambda \approx c_2 e^{3.4|\lambda|}$ (Last-100 Adding) indeed shows high-magnitude eigenvalues are more important in the long-range task variation.

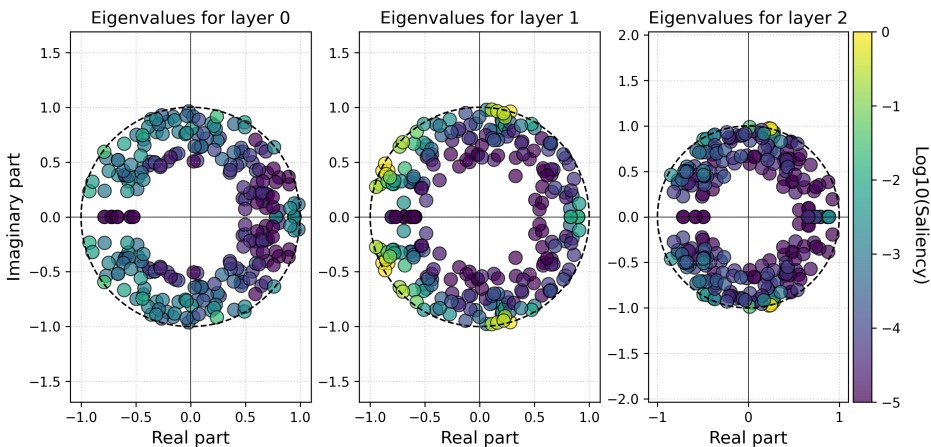

Figure 6: Parameter saliency versus eigenvalue across layers for the **First-100 Adding** task.

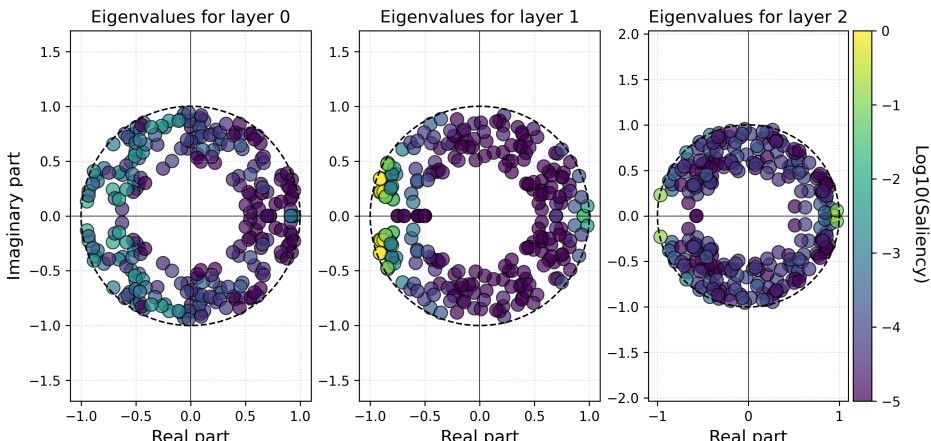

Figure 7: Parameter saliency versus eigenvalue across layers for the **Last-100 Adding** task.

The above D-LinOSS instances are also compared to another set of instances trained with eigenvalue magnitudes initialized in [0.9, 1.0]. Other hyperparameters are kept the same and aggregate test-set performance statistics across 5 random seeds are reported in Table 7 below.

Table 7: D-LinOSS on Position-Skewed Adding Task Variations.

| | First-100 Adding | | Last-100 Adding | |
|---|---|---|---|---|
| Range $|\lambda|$ | Avg. MSE | Min. MSE | Avg. MSE | Min. MSE |
| [0.5, 1.0] | 7.6e-2 | 1.1e-3 | 1.9e-4 | 1.5e-4 |
| [0.9, 1.0] | 4.4e-2 | 4.7e-4 | 2.4e-4 | 1.5e-4 |

On the First-100 Adding task, the average MSE for D-LinOSS initialized with $|\lambda| \in [0.9, 1.0]$ is $1.7\times$ lower than with $|\lambda| \in [0.5, 1.0]$, and the best in-class MSE is $2.3\times$ lower. For the Last-100 Adding task, lower damping doesn't appear to offer any empirical benefit over the larger range of initialized eigenvalues, and in fact it performs worse on average by a factor of $1.3\times$.

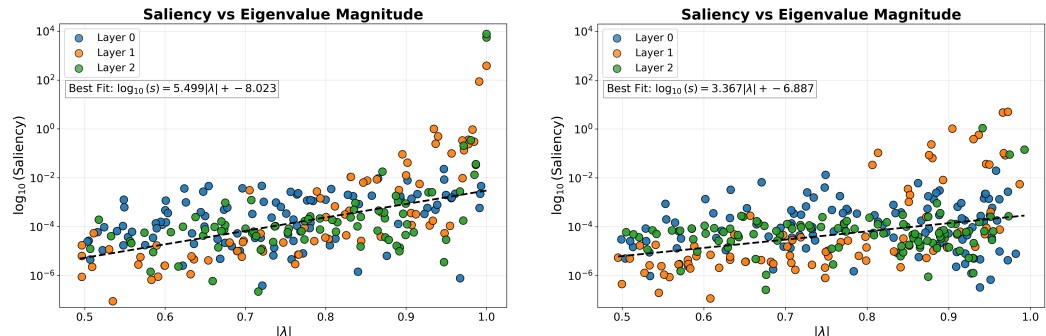

Figure 8: Parameter saliency versus eigenvalue magnitude for the **First-100 Adding** task (Left) and **Last-100 Adding** task (Right)

## C.2 STATIC DAMPING BASELINE

A natural question to ask about the proposed energy dissipation mechanism is whether or not a static amount of system damping suffices for expressive model performance. "Static" damping is defined as a constant value of $\mathbf{G}$ which is not updated throughout the training loop. We develop this baseline and compare performance to D-LinOSS on the classification and regression tasks of Tables 2 and 3.

For each dataset, the top-performing D-LinOSS hyperparameter configurations from Table 5 are re-used in the statically damped model variations. Damping values are set identically to $\mathbf{G} = 1$ and held constant through training. Model instances are run across the same 5 random seeds and average test-set metrics are reported below.

Table 8: D-LinOSS compared to a static damping baseline using the same hyperparameters. Test accuracies averaged over five different seeds on the UEA classification tasks and PPG regression task.

| | Worms ↑ | SCP1 ↑ | SCP2 ↑ | Ethanol ↑ | Heartbeat ↑ | Motor ↑ | PPG $\times 10^{-2}$ ↓ |
|---|---|---|---|---|---|---|---|
| Variable $\mathbf{G}$ | $93.9 \pm 3.2$ | $88.9 \pm 3.0$ | $58.6 \pm 2.3$ | $29.9 \pm 0.6$ | $75.8 \pm 4.9$ | $61.1 \pm 2.0$ | $6.16 \pm 0.73$ |
| Static $\mathbf{G}$ | $85.6 \pm 5.3$ | $85.7 \pm 6.5$ | $51.9 \pm 8.4$ | $26.1 \pm 3.8$ | $69.7 \pm 3.1$ | $53.7 \pm 7.7$ | $12.36 \pm 0.49$ |

The statically damped model instances perform significantly worse than D-LinOSS. Although the full set of hyperparameter sweeps was not run for the baseline, which would be a fairer comparison, the poor performance indicates that statically defined damping is not a robust parameterization within the D-LinOSS framework, compared to adaptively learning distributions of $\mathbf{G}$. In fact, the unassuming selection of $\mathbf{G} = 1$ appears to be an adversarial choice of constant damping, as this baseline performs worse than the identically undamped variant LinOSS-IMEX. In conclusion, the eigenvalue distributions and underlying gradient landscape of parameters $\mathbf{G}$, $\mathbf{A}$, and $\Delta t$ are a sensitive and critical aspect of these state-space models that require principled and flexible mechanisms, such as learnable damping terms and properly tuned initialization schemes.

## C.3 PPG CONVERGENCE

We show the model convergence of D-LinOSS, LinOSS-IM, and LinOSS-IMEX on the PPG-DaLiA long-range regression experiment. Additional randomly seeded model instances of the top-performing hyperparameter configurations from Table 5 and the corresponding table in Rusch & Rus (2025) are trained, and the validation metrics of the best performing model instances are shown in Figure 9. D-LinOSS converges to a validation metric of 7.5e-2 2.7× faster than LinOSS-IM and 4.6× faster than LinOSS-IMEX. Complementing the convergence results on the synthetic task of Section 4.4, we see that D-LinOSS converges faster on real-world datasets as well.

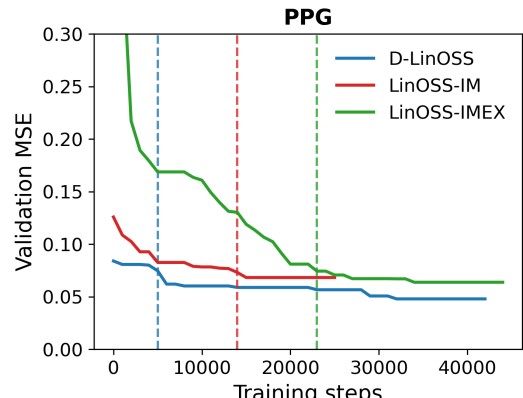

Figure 9: Validation metric convergence for top-performing model configurations on **PPG-DaLiA**.

# D    GLOSSARY

## D.1    ENERGY DISSIPATION, FORGETTING, AND DAMPING

In this paper, the terms "energy dissipation," "forgetting," and "damping" are used frequently and somewhat interchangeably. We provide a more formal description of these terms below.

- **Energy dissipation** refers to the time-evolution of the norm $||\cdot||$ of the SSM layers' internal state vector, denoted $\mathbf{x} \in \mathbb{R}^m$. This state vector norm can be interpreted as the "energy" or "informational content" held by the recurrence at any given time step. When viewing the recurrence as a decoupled second-order system expressed in some eigen-basis, we see that the dissipation of this system energy is due to the exponentiation of eigenvalues with magnitude less than 1.

$$\mathbf{x}_n = \mathbf{M}^n \mathbf{x}_0 = \lambda^n \mathbf{x}_0$$
$$||\mathbf{x}_n|| = |\lambda|^n ||\mathbf{x}_0||$$
$$|\lambda| < 1 \ \Rightarrow \ \lim_{n \to \infty} ||\mathbf{x}_n|| = 0$$

  The amount of energy dissipation is equivalently interpreted as how far below 1 the eigenvalue magnitudes are, i.e. how quickly the above limit converges.

- **Forgetting** is used equivalently to energy dissipation, but may reflect a more abstract interpretation of shedding informational content contained in the recurrent state vector $\mathbf{x}$.

- **Damping** refers to the underlying continuous-time parameter $\mathbf{G}$ that D-LinOSS layers are discretized from. As there is a bijective relationship between parameters and eigenvalues, and further there is an inversely proportional relationship between $\mathbf{G}_i$ and $|\lambda_i|$, more system damping (higher $\mathbf{G}$) can be interpreted as more energy dissipation, and less system damping (lower $\mathbf{G}$) can be interpreted as more energy retention.

$$|\lambda_i| = \frac{1}{\sqrt{1 + \Delta t_i \mathbf{G}_i}}$$

