# OpenReview forum: "Learning to Dissipate Energy in Oscillatory State-Space Models"
_ICLR.cc/2026/Conference — Submitted to ICLR 2026_

### Official Review · Reviewer_h2wS · 2025-10-21

**Soundness:** 2
**Presentation:** 2
**Contribution:** 2
**Rating:** 4
**Confidence:** 3

**Summary:**

This paper introduces the Damped Linear Oscillatory State-Space Model (D-LinOSS), an extension of the Linear Oscillatory State-Space Model (LinOSS), which models sequence data through layers of discretized forced harmonic oscillators. While previous LinOSS models coupled frequency and damping, restricting energy dissipation to a single time scale, D-LinOSS introduces learnable damping parameters that allow the system to adaptively dissipate energy on arbitrary time scales.

**Strengths:**

This paper presents a solid and well-motivated contribution to the theory and design of oscillatory state-space models. By introducing learnable damping into the Linear Oscillatory State-Space Model (LinOSS), the proposed D-LinOSS enhances the model’s ability to control forgetting and energy dissipation over arbitrary time scales. This addresses a key limitation of previous LinOSS variants, where frequency and damping were rigidly coupled, restricting flexibility in long-range dynamics. The work is theoretically rigorous, providing a detailed spectral analysis and stability proof that clarifies how the new parameterization expands the set of representable stable systems.

On the empirical side, the paper supports its theoretical insights with well-designed numerical experiments on synthetic and real-world datasets such as PPG-DaLiA and the Weather forecasting benchmark. These results, though modest in scale, consistently demonstrate that D-LinOSS achieves faster convergence, improved stability, and slightly higher accuracy compared to prior oscillatory SSMs. Moreover, the authors’ discussion of universality—showing that D-LinOSS preserves the universal approximation property of LinOSS—adds conceptual depth and connects the work to broader theoretical frameworks in sequence modeling. Overall, the paper offers a meaningful and well-executed advancement in understanding and improving the forgetting mechanisms of state-space models.

**Weaknesses:**

The main limitation of the paper lies in its empirical evaluation, which appears relatively underdeveloped given the strength of its theoretical contributions. The experimental results seem under-trained and insufficiently tuned, particularly for the baseline comparisons—there is little evidence that the baselines were optimized to their best performance. This makes it difficult to assess the true effectiveness of the proposed model.

Furthermore, the study is limited to small-scale numerical experiments, leaving open the question of whether the claimed properties—especially those related to stability and forgetting—would generalize to larger or more complex datasets. Without experiments at scale or more diverse benchmarks, the practical impact of the proposed approach remains uncertain. To strengthen the paper, the authors could include carefully tuned baselines, ablation studies, and larger-scale experiments that better validate the theoretical claims in real-world scenarios.

**Questions:**

Can you provide the detail about the tuning of the numerical experiments provided in the paper?

---

> ### Author Response · Authors · 2025-11-22
> **Author Response**
>
> We thank the reviewer for their careful reading of our article, and we are glad to hear the reviewer found our work "well-motivated," "theoretically rigorous," following "well-designed numerical experiments on synthetic and real-world datasets," and overall a "well-executed advancement in understanding."  We address the reviewer's main concern about the empirical results below, clarifying the experimental setup.
>
> *The experimental results seem under-trained and insufficiently tuned, particularly for the baseline comparisons—there is little evidence that the baselines were optimized to their best performance... Can you provide the detail about the tuning of the numerical experiments provided in the paper?*
>
> We thank the reviewer for the opportunity to clarify the experimental setup followed in the paper.  The experimental design is precisely replicated from [1], which involves extensive hyperparameter tuning and randomized repeats to ensure robustness in results.  In particular, for each dataset considered in the six UEA-MTSCA classification tasks and the regression task of PPG-DaLiA, each model is trained on a grid of 162 pre-defined hyperparameters, and 5 random seeds are run for each configuration.  In total, just for evaluating D-LinOSS on these 7 datasets, a total of 5,760 model instances were trained from scratch, offering a very consistent and robust comparison to the other baseline models.  Results for the Weather dataset instead keep consistent with the baselines provided in [2] to maintain fairness, which are found through a random hyperparameter search space without seeded repeats.
>
> *To strengthen the paper, the authors could include carefully tuned baselines, ablation studies, and larger-scale experiments that better validate the theoretical claims in real-world scenarios.*
>
> We hope that in addition to the eight real-world datasets provided in the empirical results section, the reviewer finds the new ablations in Appendix C of interest, adding insight to D-LinOSS's learnable energy dissipation mechanism.  Specifically, in Appendix C.1, we conduct a new parameter sensitivity analysis motivated by theory in classical neural network compression methods, investigating how D-LinOSS learns to ``activate" recurrent modes with high-magnitude eigenvalues more strongly in long-range learning scenarios.  Appendix C.2 provides an additional baseline, comparing the adaptive, learnable damping mechanism to a simplified, statically defined approach.  Appendix C.3 builds off of the convergence results from Section 4.4, showing that D-LinOSS boasts similar benefits on real-world datasets.
>
> We hope that we have clarified the experimental design behind the numerical results and added insight to the D-LinOSS framework in the revised version of the paper.  We would kindly ask the reviewer to update their score accordingly.
>
> [1] Benjamin Walker, Andrew D. McLeod, Tiexin Qin, Yichuan Cheng, Haoliang Li, and Terry Lyons.
> Log neural controlled differential equations: The lie brackets make a difference. International
> Conference on Machine Learning, 2024.
>
> [2] Albert Gu, Karan Goel, and Christopher Ré. Efficiently modeling long sequences with structured state spaces. arXiv preprint arXiv:2111.00396, 2021.

---

### Official Review · Reviewer_nwX2 · 2025-10-31

**Soundness:** 2
**Presentation:** 3
**Contribution:** 1
**Rating:** 4
**Confidence:** 3

**Summary:**

This work revisits a very well celebrated work from last year and identifies a shortcoming in the set of equations describing the linear oscillatory state space model (LinOSS). Namely, the set of equations did not involve a damping term, which the authors in this work introduce. They show that with the addition of this term enables faster, efficient, and effective training across various benchmarks.

**Strengths:**

- The addition of the dampening term is a trivial, yet needed, addition that is well motivated and, in this reviewer's opinion, should have been part of the LinOSS to begin with. In this sense, authors have identified an important gap.

- Benchmarks are comprehensive and convincing that the addition of the dampening term is broadly useful; though more is needed to support the specific claims (see below).

**Weaknesses:**

- The scientific novelty/contribution is quite limited. The main model, i.e. LinOSS, in my opinion is a very important direction that future research should build on. However, the addition here is trivial, and does not necessarily lead to new novel insights. To me, this paper feels like a comment on the original paper as opposed to having enough novelty for deserving a paper of its own.

- I find the claim "reducing the hyperparameter search space by 50%" to be slightly misleading. Practically, one should have just simulated an oscillatory model as accurately as possible. In that sense, the choice between implicit or implicit-explicit integration method should not have been presented as an hyperparameter by the original work, since the former is effectively introducing decay that does not exist in real continuous dynamics. In my opinion, the fact that explicitly modeling decay corrects for this "spurious capability" resulting from incorrect discretization is a much more interesting contribution as opposed to "taking away" one hyperparameter. Thus, I would like to recommend emphasizing this contribution to the scientific methodology as opposed to reduction in hyperparameter count.

**Questions:**

- Could you please use the same symbols as the LinOSS paper for Eq. 1? It seems the variables x and y are switched, which leads to confusion on the part of the reader.

- It seems in Tables 2 and 3 that LinOSS-Im is more or less at the same level as D-LinOSS? The same may be true for Table 4, though no error bars are reported (for a good reason due to data scarcity, so not authors' fault...).  In that sense, can a devil's advocate say learning to decay is not that important as long as some amount of decay is built-in? For instance, could you please add a baseline in which the decay terms are set equal to some value (defined by the task requirements, or some hyperparameter)? Will D-LinOSS still beat this baseline?

- Looking at Figure 2, it seems that increasing the sequence length makes D-LinOSS more or less the same accuracy and speed as LinOSS-Im? Can you show training for longer sequences? Moreover, similar to above, can you fix the G entries and add this new model as a baseline too?

Overall, my current assessment is as follows:  On the empirical side, more experiments are needed to fully understand when learning the decay timescales is necessary. Theoretically, this work makes an important addition to a very influential model that was given an oral presentation in last year's ICLR conference. However, this addition almost feels like a correction, as opposed to introducing a new capability of the model. In that sense, what makes this model very exciting is already published and very well celebrated by the community. In my assessment, this work could be published as a comment on the original work, for instance in TMLR; but does not satisfy the significance and novelty requirements of ICLR.

---

> ### Author Response · Authors · 2025-11-22
> **Author Response**
>
> We thank the reviewer for their thoughtful consideration of our paper and for their constructive feedback.  We appreciate that the reviewer found our work "well-motivated" and "identifies an important gap," with "benchmarks [that] are comprehensive and convincing," containing an "interesting contribution...to the scientific methodology".  Below, we address the questions and concerns raised by the reviewer.
>
> *It seems the variables x and y are switched, which leads to confusion on the part of the reader.*
>
> We thank the reviewer for catching this, the equations have been updated accordingly.
>
> *Can a devil's advocate say learning to decay is not that important as long as some amount of decay is built-in?  For instance, could you please add a baseline in which the decay terms are set equal to some value (defined by the task requirements, or some hyperparameter)?*
>
> The reviewer raises a very interesting point on whether or not a simpler energy dissipation mechanism, such as setting a constant amount of damping, can also be an effective approach.  To investigate this, we develop a baseline model with statically set damping in Appendix C.2 and compare the performance to D-LinOSS on seven of the datasets.  Notably, the baseline performs quite poorly; although the resulting baseline state-space model is also an oscillatory state-space model with ``some amount of decay," the performance is severely limited.  Indeed, the eigenvalue distributions and underlying gradient landscape are a critical aspect of oscillatory SSMs that requires a more principled approach, such as adaptive/learnable damping terms and a suitably developed initialization scheme.
>
> *Looking at Figure 2, it seems that increasing the sequence length makes D-LinOSS more or less the same accuracy and speed as LinOSS-Im? Can you show training for longer sequences?*
>
> Additional experiments show that all considered models fail to learn a correct solution for the Adding task of longer sequence lengths (in particular, T=10,000 and T=20,000).  Beyond Adding, we also provide the best-performing model convergence times on the long-range PPG-DaLiA regression task in Appendix C.3, which offers a similar takeaway: D-LinOSS reaches lower MSE significantly faster than the previous LinOSS methods.
>
> *...the addition here is trivial, and does not necessarily lead to new novel insights.*
>
> Deriving D-LinOSS instead from systems of damped linear oscillators is indeed a simple motivation, but the resulting state-space model is new, with different theoretical properties, consistently better empirical performance, and meaningful practical benefits.  Additionally, the simplicity of the proposed method can be considered a strength, as the aforementioned benefits come without any significant cost in model complexity.  Synthetic experiments studying exponential decay and convergence time provide meaningful insight to how SSM spectral properties generally impact learning performance in different scenarios.  Complementing these results, we hope the reviewer finds interest in the additional experiments performed in Appendix C.1, which provide more interpretation of the learned weights and insight into the underlying learnable damping mechanism.  In particular, this ablation demonstrates that learned parameters associated with higher-magnitude system eigenvalues are more important (sensitive) in tasks that require capturing longer-range dependencies.
>
> *I find the claim "reducing the hyperparameter search space by 50%" to be slightly misleading.*
>
> We apologize for the confusion.  We have adjusted phrasing in the work to highlight that D-LinOSS relinquishes the use of multiple discretization schemes, which is used as a binary hyperparameter in prior LinOSS methods.  As these two discretization schemes were originally used to endow the resulting state-space model with different spectral properties (empirically shown to be useful in different scenarios) D-LinOSS is a more general framework that can capture all stable second-order systems, moving past the need for this hyperparameter.
>
> We sincerely hope that we have addressed the concerns of the reviewer satisfactorily in the revised version and would kindly ask the reviewer to update their score accordingly.

---

> ### Comment · Reviewer_nwX2 · 2025-11-25
>
> I truly appreciated the author responses. Unfortunately, as noted in my original review, my main concern is that the premise of the paper does not clear the bar for acceptance at a top venue as a follow-up to the last year's oral presentation. Addition of a decay term does not constitute a major improvement in novelty, which is arguably a subjective area of assessment, but I am being asked to assess it anyways in the reviewer guidelines. Taking that criteria out, I do believe I would have accepted the revised manuscript on the spot for TMLR, a venue that welcomes incremental improvements on published algorithms. I think the presented results are solid and no major technical concern is there, and there was not much to begin with.
>
> AC may not agree with me and can choose to accept, but this is my honest evaluation. To be more fair to authors, I will also note that my main research track is more theoretical and I may not correctly appreciate the practical contributions.

---

### Official Review · Reviewer_Uv1a · 2025-10-31

**Soundness:** 3
**Presentation:** 2
**Contribution:** 3
**Rating:** 6
**Confidence:** 4

**Summary:**

The paper introduces a new continuous-time recurrent network by extending Linear Oscillatory State-Space models (LinOSS). Specifically, the proposed Damped LinOSS (D-LinOSS) adds a damping term to the continous-time forced second-order ODEs underlying each layer. This damping term enables learnable energy dissipation, allowing the discrete-time model to represent a wider range of stable dynamics than previous discretzied LinOSS models. On average, D-LinOSS outperforms LinOSS (both the IM and IMEX discretizations) and reaches State-of-the-Art performance on the evaluated benchmark tasks.

**Strengths:**

- The empirical results of D-LinOSS on several real-world datasets are significant, improving State-of-the-Art as well as the previous LinOSS models.
- The addition of damping to D-LinOSS offers clear practical and theoretical advantages over LinOSS models, enabling eigenvalues that cover the entire unit disc, thus uncoupling frequency and energy dissipation. Table 1 and Figure 3 are particularly compelling illustrations of these advantages.

**Weaknesses:**

- Please clearly define (and if necessary, differentiate) the terms: energy dissipation, forgetting, damping, and their relation to real and imaginary eigenvalue components, eigenvalue magnitude, and oscillation frequency. Particularly energy dissipation, damping, and forgetting are often used loosely and somewhat interchangeably, which can be confusing. It may help to add a section defining important vocabulary at the beginning of Section 2.
- The proof of bijectivity in Proposition A.3 is not sufficiently elaborated. Particularly, more details on the exact derivation of the inverse function from eigenvalues to A, G would help convince that it is the true inverse. In addition, classical injectivity-surjectivity arguments would aid in comprehension.
- Figure 1 is somewhat difficult to parse. Using unnormalized time and different colors for each “frequency” as well as clearly labelling the z axis could go a long way. In addition, the caption should more closely explain the figure and directly reference it. Finally, it might be worth considering including Figure 3 in Figure 1 (or at least more prominently placing Figure 3 since it’s quite easy to understand and clarifies the main contribution of adding learnable damping).
- The phrase "reducing the hyperparameter search space by 50%” is a bit misleading (unclear whether you are referring to the number of hyperparameters which are eliminated or the entire search space covered by all hyperparameters...). Maybe rephrase this to clearly state that you are just eliminating the need to choose between LinOSS-IM and LinOSS-IMEX.
- Even though the empirical results are strong, their interpretation is missing. Please elaborate on possible reasons why D-LinOSS is better at the UEA Motor task while lagging on the Heartbeat task.

If these weaknesses are sufficiently addressed, I am happy to increase my rating score.

**Questions:**

- What is meant by the trainability of the timestep parameters in line 124? In the original LinOSS, the timestep is not trainable and fixed.
- Could you add some more intuition on why LinOSS-IMEX and LinOSS-IM can be universal while still failing to model exponential decay?
- One of the main reasons to choose LinOSS-IMEX over LinOSS-IM is that it can model conservative systems. How does D-LinOSS compare to LinOSS-IMEX when modelling conservative systems? What solution does D-LinOSS reach (i.e., does it actually learn to set G=0, or does it find some other solution for conservative systems)?
- Figure 3 gives a clear explanation of why D-LinOSS is better than LinOSS, theoretically. If we plot the actual eigenvalues after training on a real-world task (of your choice), is this behavior verified? This would give more body to the answer to “A natural question is whether or not a larger set of reachable eigenvalues is empirically useful” of line 250.
- Figure 2 confirms the faster convergence on the synthetic task. Is this benefit also observed on the real-world datasets? A similar figure in the appendix would be a great addition.

---

> ### Author Response · Authors · 2025-11-22
> **Author Response (1/2)**
>
> We thank the reviewer for their careful and attentive reading of our paper and for their constructive feedback.  We are glad the reviewer appreciates the paper's "significant empirical results" and the proposed method's "compelling advantages".  Below, we address the reviewer's questions and concerns.
>
> *Please clearly define (and if necessary, differentiate) the terms: energy dissipation, forgetting, damping, and their relation to real and imaginary eigenvalue components, eigenvalue magnitude, and oscillation frequency.*
>
> The use of these terms is now clarified in Appendix D.1, providing context on how they relate to state vector norms, eigenvalue magnitudes, and the underlying D-LinOSS parameters.
>
> *The proof of bijectivity in Proposition A.3 is not sufficiently elaborated.*
>
> We thank the reviewer for their detailed reading of the supplementary materials, which has allowed us to make improvements in the correctness and clarity of the arguments.  The proof of bijectivity has been re-written in Appendix A.3, including more detail, verifying that the proposed function is indeed a well-defined inverse, and making explicit connections to injectivity and surjectivity.  Additionally, we have made the correction that the image of the bijection does not include 0 eigenvalues, which was overlooked in the previous manuscript.
>
> *Figure 1 is somewhat difficult to parse.*
>
> The figure has been adjusted to include z-axis labels and the caption now more directly describes what is depicted in the figure.  The normalization of time on the x-axis remains a necessary part of the figure: although there is a 1-1 coupling between frequency and magnitude for previous LinOSS models, in the case of LinOSS-IM the one-dimensional curve does not lie on the unit ring $|\lambda| = 1$ (meaning damping is not identical across frequency), so plotting the oscillatory behavior over normalized time captures the degeneracy in expressivity in the general case.  Additionally, Figure 3 has been placed in a more prominent section in the paper, as suggested.
>
> *The phrase "reducing the hyperparameter search space by 50%” is a bit misleading... Maybe rephrase this to clearly state that you are just eliminating the need to choose between LinOSS-IM and LinOSS-IMEX.*
>
> We apologize for the confusion; we simply mean D-LinOSS relinquishes the need for multiple discretization schemes, which was a binary hyperparameter included in the search space of previous LinOSS methods.  The language regarding this has been rephrased throughout the paper accordingly.
>
> *Even though the empirical results are strong, their interpretation is missing. Please elaborate on possible reasons why D-LinOSS is better at the UEA Motor task while lagging on the Heartbeat task.*
>
> The UEA classification datasets are quite small in size, meaning there is non-negligible variability in performance results.  The experimental design aims to combat this by using a consistent hyperparameter search space (of 162 configurations per model) and repeating model instances across 5 seeds, but there is still some noise to be expected when looking at any one dataset.  The most statistically significant result is the average across all 6 datasets, which shows improvement over both LinOSS-IM and LinOSS-IMEX.  For further interpretation of the performance on MotorImagery, as well as broad interpretation of the underlying D-LinOSS mechanism, the reviewer may be interested in looking at the learned eigenvalue distributions shown in Appendix B.2 or the new parameter saliency analysis conducted in Appendix C.1.

---

> ### Author Response · Authors · 2025-11-22
> **Author Response (2/2)**
>
> *What is meant by the trainability of the timestep parameters in line 124? In the original LinOSS, the timestep is not trainable and fixed.*
>
> Timestep parameters $\Delta t$ are adaptively learned between (0, 1.0] through a sigmoid activation parameterization, governing the integration timestep of the discrete-time approximation of the continuous-time dynamics.  This mechanism is adopted directly from the original LinOSS model, which indeed leverages trainable timestep parameters as well.
>
> *Could you add some more intuition on why LinOSS-IMEX and LinOSS-IM can be universal while still failing to model exponential decay?*
>
> D-LinOSS achieves orders of magnitude better performance on the synthetic experiment, but previous LinOSS models still achieve reasonably low MSE.  While LinOSS can still successfully approximate temporal relationships not directly in their range of reachable eigenvalues, the expressivity limitations become more evident when compared to D-LinOSS.
>
> *How does D-LinOSS compare to LinOSS-IMEX when modelling conservative systems? What solution does D-LinOSS reach (i.e., does it actually learn to set G=0, or does it find some other solution for conservative systems)?*
>
> We first note that eigenvalue distributions in learned model instances appear to stay roughly distributed according to their initialization.  Further, we conduct a new parameter "saliency" (sensitivity) analysis in Appendix C.1 to study how eigenvalues with different amounts of damping (G values) impact long-range learning; it is observed that eigenvalues with higher magnitude (equivalently, lower G) are more important to the loss function in the learned model solution.  Motivated by these observations, we offer the interpretation that D-LinOSS learns to ``activate" different recurrent modalities that are already available, rather than shift eigenvalues over far distances to capture new dynamics.  As such, D-LinOSS would likely behave similarly for conservative systems, activating eigenvalues with $G \approx 0$.
>
> *Figure 3 gives a clear explanation of why D-LinOSS is better than LinOSS, theoretically. If we plot the actual eigenvalues after training on a real-world task (of your choice), is this behavior verified?*
>
> Learned eigenvalue distributions for the MotorImagery task are plotted in Appendix B.2, which qualitatively stay aligned with the initialized distributions.  Recall that special initialization schemes have been developed for each model to place eigenvalues in a suitable sub-region of the reachable eigenvalue ranges shown in Figure 3.
>
> *Figure 2 confirms the faster convergence on the synthetic task. Is this benefit also observed on the real-world datasets?*
>
> Best-performing model convergence for the long-range regression experiment on PPG-DaLiA is provided in Appendix C.3, demonstrating similar gains in convergence speed on this real-world dataset.
>
> We sincerely hope that we have addressed the concerns of the reviewer satisfactorily in the revised version and would kindly ask the reviewer to update their score accordingly.

---

### Official Review · Reviewer_F45N · 2025-11-06

**Soundness:** 4
**Presentation:** 4
**Contribution:** 2
**Rating:** 6
**Confidence:** 3

**Summary:**

In their paper, the authors build upon LinOSS and extend the previous physics-inspired oscillatory state-space model by introducing learnable damping. They identify limitations in existing oscillatory SSMs and theoretically motivate learnable damping. The authors perform a small synthetic experiment to learn exponentially decaying functions, and show that D-LinOSS outperforms LinOSS across eight different real-world learning tasks. The authors derive theoretical proofs of the improved representational capacity of D-LinOSS, while maintaining computational efficiency.

**Strengths:**

- The authors clearly outline the contributions of their paper, highlighting the limitations and improvements over existing oscillatory state-space models (LinOSS).
- The authors conduct rigorous theoretical analysis of their method and provide a link between the new parameterization to improved representational capacity. They prove that D-LinOSS layers span the full unit disk in the complex plane.
- The authors provide a controlled synthetic experiment (learning exponential decay) to justify D-LinOSS and an empirical evaluation on more complex, real-world learning tasks.
- The authors present thorough empirical results outperforming LinOSS across eight datasets with details on hyperparameters, good reproducibility.
- Figure 1 is very well-designed.

**Weaknesses:**

- Most reported improvements in the presented empirical results are marginal. I would like to see more ablation on model size/parameter count to be sure of the practical significance of D-LinOSS.
- While D-LinOSS is well-motivated, the contribution feels like a simple extension of LinOSS. The contribution feels incremental in the broader landscape without additional analysis on the interpretation of learned weights.

**Questions:**

Are the learned G values interpretable? Do there exist any links to the underlying physical motivation? Can you show qualitative evidence that learned damping captures long-term versus short-term dependencies?

---

> ### Author Response · Authors · 2025-11-22
> **Author Response**
>
> We thank the reviewer for their thoughtful consideration of our work, and we are glad the reviewer finds the paper "clear", "rigorous", and "thorough".  Below, we address the reviewer's questions.
>
> *The contribution feels incremental in the broader landscape without additional analysis on the interpretation of learned weights... Are the learned G values interpretable? Do there exist any links to the underlying physical motivation? Can you show qualitative evidence that learned damping captures long-term versus short-term dependencies?*
>
> We have included a new section in Appendix C.1 to analyze the learned damping parameters (and their corresponding eigenvalues) and interpret their relation to learning long-term dependencies.  In particular, we develop long-range and short-range variations of the synthetic Adding task, and we conduct a parameter ``saliency" (sensitivity) analysis motivated by theory in classical neural network compression.  In short, we see that for tasks requiring long-range learning, the loss function is more sensitive to eigenvalues with higher magnitude (equivalently, smaller G value).  In contrast, for tasks that do not require long-range learning, there is a less pronounced relationship between G value and parameter importance.  Complementing this, we observe stronger raw performance on the long-range Adding task for D-LinOSS instances initialized with lower amounts of damping.  These results indicate that lower amounts of system damping are more useful for capturing long-term dependencies, matching our intuition for how the SSM's underlying energy dissipation mechanism impacts learning.
>
> Additional depictions of the learned eigenvalue distributions on the MotorImagery for the different models can be seen in Appendix B.2.
>
> *I would like to see more ablation on model size/parameter count to be sure of the practical significance of D-LinOSS.*
>
> Comparing the parameter counts and run times of best-performing model instances in Table 6 of Appendix B.5 does not show any apparent relationship between model size and performance within the current suite of experiments.  Characterizing the scaling laws of LinOSS-type models remains an area of future work; currently, the observable practical significance of D-LinOSS lies in the consistent performance improvements, faster convergence, interpretable mechanism, and smaller hyperparameter space.
>
> We hope the additional analysis on parameter sensitivity has increased the interpretation of the proposed learnable damping mechanism and improved the value of our work.  We would kindly ask the reviewer to update their score accordingly.

---

### Author Response · Authors · 2025-12-03
**Summary of Rebuttal Developments**

### __Recognition of Circumstances__
We apologize to the AC for the extenuating circumstances and offer our sincere appreciation for the extra time and effort spent providing fair review in the challenging situation.  To help streamline the process, we summarize developments from the rebuttal phase below.

### __Paper Summary__
We present a new deep state-space model, D-LinOSS, inspired from second-order LTI dynamical systems.  Generalizing the LinOSS framework, the resulting SSM has strict theoretical improvements in expressive capacity, achieves consistently stronger performance on real-world datasets, converges faster, and reduces the hyperparameter space, all without introducing model complexity.  A robust experimental results section spanning 10 datasets and 16 baselines complements theoretical proof and interpretive ablations, providing insight on both the practical utility and internal mechanics of learning dissipative terms in oscillatory state-space models.

### __Improvements__
In response to reviewer feedback, additional experiments and revision made during the rebuttal phase have made significant improvement to the paper.  We list these contributions below.

### A. Interpretation of Parameterization
#### Reviewers: F45N, Uv1a
Reviewers brought to our attention that the model damping mechanism could use more interpretation and intuitive understanding.  To address this, we have conducted a parameter sensitivity analysis, highlighting the crucial relationship between damping terms G, eigenvalue magnitude, and loss function sensitivity in both long and short-range learning scenarios.  We also include illustrations of empirically observed eigenvalue distributions on one of the real-world datasets to elucidate the difference in underlying model dynamics.

### B. Necessity of Model Design
#### Reviewers: nwX2
A primary question was whether or not the specific parameterization of D-LinOSS’s learnable energy dissipation mechanism is strictly necessary — could a simpler technique be used?  We develop a new baseline state-space model which incorporates a static amount of decay, and show this model greatly underperforms the adaptive and principled approach we propose in the paper.

### C. Extending Convergence Results
#### Reviewers: Uv1a, nwX2
Reviewers were interested if the faster convergence behavior also holds true for real-world datasets.  To answer this question, we extend our investigation to PPG-DaLiA and show similar results on improved convergence time, highlighting D-LinOSS’s consistently faster learning behavior.

### D. Theoretical Clarifications
#### Reviewers: Uv1a, nwX2
We clarify points of confusion and revise the theoretical discussion for enhanced understanding.  In particular, we expand the proof A.3 in great detail.  We provide in-depth descriptions of frequently used terms, fix notational miscommunications, and re-phrase discussion on the reduction of model hyperparameter space.

### __Exchange with Reviewers__
The direct outcomes of our revisions:
1. We have included an in-depth ablation to answer Reviewer F45N’s main question on model interpretability.
2. We address every point raised by Reviewer Uv1a, through additional experiments and discussion, who expressed they would be “happy to increase [their] rating score”.
3. Although Reviewer nwX2 considers our work an incremental contribution, in response to our rebuttal, they expressed they “would have accepted the revised manuscript on the spot for TMLR…the presented results are solid.”

### __AI-Generated Review__
We bring to the attention of the AC that review h2wS lacks factual accuracy regarding the work’s empirical results section, and it is seen that this review is fully AI-generated.  (https://iclr.pangram.com/reviews?submission_number=22097).  The only criticism raised is “experimental results seem under-trained and insufficiently tuned, particularly for the baseline comparisons—there is little evidence that the baselines were optimized to their best performance.”  This is false; baselines are fine-tuned on randomized sweeps across large, controlled hyperparameter search spaces on each dataset, ensuring fair and consistent comparison.  These baselines are sourced from [1] and [2], and the experimental design from these works is precisely followed when evaluating our model.  We thus follow guidelines from the PC: “authors who receive very poor quality or LLM-generated reviews should flag them to their ACs.”

### __Thank You__
We thank the reviewers for their detailed reading of our paper and constructive feedback which has helped us make improvements to our work.  We thank the AC for their time, effort, and consideration in this process.

[1] Benjamin Walker, Andrew D. McLeod, Tiexin Qin, Yichuan Cheng, Haoliang Li, and Terry Lyons. Log neural controlled differential equations: The lie brackets make a difference.

[2] Albert Gu, Karan Goel, and Christopher Ré. Efficiently modeling long sequences with structured state spaces.

---

### Meta-Review · Area_Chair_sn34 · 2025-12-07

**Summary:**

This paper proposes D-LinOSS, a damped extension of the Linear Oscillatory State-Space Model (LinOSS) for continuous-time recurrent architectures. More specifically, the previous LinOSS models lacked learnable damping, limiting their ability to represent stable and diverse temporal dynamics. D-LinOSS introduces independent damping parameters into the underlying second-order ODEs, enabling adaptive energy dissipation and improving representational capacity while preserving computational efficiency. The authors provide theoretical justification for the increased expressivity and perform a synthetic experiment demonstrating superior learning of exponentially decaying functions. Across eight real-world sequence modeling benchmarks, D-LinOSS consistently outperforms LinOSS and achieves state-of-the-art performance.

The proposed algorithm is a damped extension of Rusch & Rus (2025), and thus the novelty appears incremental. Although the authors validate their model on several real-world tasks, the performance gains are not particularly substantial. While reviewers acknowledge the value of the theoretical contributions, incorporating more practical setups would strengthen the work. Therefore, I encourage the authors to revise the paper in accordance with the reviewers’ comments and consider resubmitting to a future venue.

**Reviewer Concerns:**

1. Novelty of the proposed method over Rusch & Rus (2025)
2. Empirical results.

It is difficult to address 1. So, the key point is 2. The performance of the proposed method for small scaled real dataset would be great. However, it would be great to have more paractical experiments where Mamba can be used.

**Reviewer Scores:**

After the rebuttal, Reviewer Uv1a might increase the score. However, I guess other would keep their original score and this is slightly below the acceptance threshold.

---

### Decision · Program_Chairs · 2026-01-26

Reject